# Systemic infection by *Candida albicans* requires FASN-α subunit induced cell wall remodeling to perturb immune response

Yajing Zhao[1,2,☯]*, Zhishan Zhou[1,2,☯], Guiyue Cai[1,2,☯], Dandan Zhang[1,2,☯], Xiaoting Yu[1,2], Dongmei Li[3], Shuixiu Li[1,2], Zhanpeng Zhang[1,2], Dongli Zhang[1,2], Jiyao Luo[1,2], Yunfeng Hu[1], Aili Gao[4]*, Hong Zhang[1,2]*

**1** Department of Dermatology, The First Affiliated Hospital of Jinan University, Guangzhou, China, **2** Institute of Mycology, Jinan University, Guangzhou, China, **3** Department of Microbiology and Immunology, Georgetown University Medical Center, Washington District of Columbia, United States of America, **4** Guangzhou Dermatology Hospital, Guangzhou, China

☯ These authors contributed equally to this work.

* zhaoyj@jnu.edu.cn (YZ); alicegao197897@163.com (AG); tzhangh@jnu.edu.cn (HZ)

## Abstract

Invasive fungal infections are a leading cause of mortality and morbidity in patients with severely impaired host defenses, while treatment options remain limited. Fatty acid synthase (FASN), the key enzyme regulating *de novo* biosynthesis of fatty acids, is crucial for the lethal infection of fungi; however, its pathogenic mechanism is still far from clear. Here, we identified the α subunit of FASN as a potential immunotherapeutic target against systemic *Candida albicans* infection. The avirulence of the encoded gene (*FAS2*)-deleted mutant in a mouse model of systemic candidiasis is not due to its fitness defects, because sufficient exogenous fatty acids in serum can overcome FASN inhibition. However, the *FAS2*-deleted mutant displays increased circulating innate immune responses and enhances activated neutrophil fungicidal activity through the unmasking of immunogenic cell wall epitopes *via* the Rho-1 dependent Mkc1-MAPK signaling pathway, which facilitates fungal clearance, reduces renal tissue damage and inflammatory cell infiltration, ultimately lowers fungal pathogenicity. Priming with the *FAS2*-deleted mutant provided significant protection against subsequent lethal infection with wild-type *C. albicans* in mice as early as one week, and it was well-tolerated with limited toxicity. Our findings indicate that the FASN-α subunit plays key roles in the regulation of neutrophil-associated antifungal immunity and could be a potential target for immunotherapeutic intervention.

## Author summary

Invasive fungal infections caused by *Candida albicans* are a significant cause of mortality in immunocompromised patients, with limited treatment options available. This study highlights the α subunit of fatty acid synthase (FASN-αsubunit), encoded by the *FAS2* gene, as a promising immunotherapeutic target. We discovered that the loss of *FAS2* boosts neutrophil activity and promotes fungal clearance by exposing fungal cell wall

**Data availability statement:** The raw single-cell RNA sequencing data generated in this study have been deposited in the NCBI's Sequence Read Archive (SRA) under the accession number PRJNA1202931. You can access it at NCBI SRA (https://www.ncbi.nlm.nih.gov/sra). The metabolomics data have been deposited in the OMIX database at the China National Center for Bioinformation / Beijing Institute of Genomics, Chinese Academy of Sciences. The accession numbers are OMIX008418 and OMIX008416. You can access the data at OMIX (http://ngdc.cncb.ac.cn/omix). All raw data used to generate the main manuscript and supplementary figures can be found in S1-S8 Data provided with this paper. All other relevant data are within the manuscript and its Supporting Information files.

**Funding:** This work was supported by the National Key Research and Development Program of China (No. 2021YFC2300400 to HZ), the National Nature Science Foundation of China (No. 82272357 to HZ, No. 81971913 to HZ, and No. 82402647 YZ), the Innovation Team Project of Guangdong University (No. 2022KCXTD002 to HZ), the Guangzhou Basic Research Program Joint Funding Project for Municipal and University/Institute Collaboration (No. 2024A03J0480 to AG), China Postdoctoral Science Foundation (No. 2022M711329 to YZ), and the Guangdong Basic and Applied Basic Research Foundation (No. 2021A1515110842 to YZ). The funders had no role in study design, data collection and analysis, decision to publish, or preparation of the manuscript.

**Competing interests:** The authors have declared that no competing interests exist.

components. This immune activation, mediated through a specific signaling pathway, reduces tissue damage and inflammation. Remarkably, the *FAS2*-deleted strain also acts as a protective immunogen, priming the host to withstand subsequent lethal infections. These findings provide new insights into the interaction between fungal metabolism and host immunity, paving the way for innovative approaches to treat systemic fungal infections.

## Introduction

Fungal infections claim an unacceptably high fatality rate, with over 6.55 million people affected by life-threatening invasive fungal disease annually and approximately 3.8 million deaths, including 2.55 million directly attributable to these infection [1]. Specifically, *Candida albicans* is one of the most common fungal pathogens in immunocompromised patients, frequently leading to life-threatening systemic conditions such as candidemia and invasive candidiasis, with a crude mortality rate of approximately 40% despite antifungal intervention [2]. This opportunistic pathogen normally exists as a commensal organism on human mucosal surfaces but becomes pathogenic in individuals with weakened immune systems, such as those undergoing chemotherapy, organ transplants, or prolonged use of corticosteroids [3]. In these patients, *C. albicans* can evade immune responses, leading to severe complications due to its ability to switch between yeast and hyphal forms and resist treatment [4].

The impaired host defenses of people suffering from fungal infections, combined with the challenges of eradicating these infections with current antifungals, highlight the importance of the immune system in determining infection outcome and prognosis of these infections. Accumulating evidence supports the notion that effective therapies against invasive fungal infections should include the advancement of immunotherapeutic strategies [5]. The objectives of immunotherapy are to increase the number of phagocytic cells and to modulate their kinetics and activity at the site of infection, thereby enabling more efficient elimination of fungal cells [6]. Several studies have shown that vaccination with attenuated pathogens, such as *Escherichia coli* [7], SARS-Cov2 [8], *Francisella tularensis* [9], and commensal intestinal fungi *Saccharomyces cerevisiae* [10], can protect the host from subsequent infections, highlighting the potential to develop immunopotentiators to combat various infectious diseases. In *C. albicans*, strategies aimed at inhibiting fungal morphological transformation can significantly reduce fungal virulence and immune escape [11]. However, despite extensive research efforts, progress in developing effective antifungal immunotherapies has lagged behind that of other clinically significant microbes.

Fatty acid synthase (FASN) catalyzes the first committed step in the *de novo* lipogenesis of fatty acid by producing saturated fatty acids from acetyl-CoA, malonyl-CoA, and NADPH, which are essential for cell membranes, energy storage, and signaling molecules [12]. Despite the conservation of the biosynthetic reaction mechanisms across the kingdom, the architecture of the FASN complexes and the organization of genes encoding the enzymes vary between vertebrates and fungi [13,14]. In mammals, FASN consists of an α2 homodimer formed from a single 270-kDa polypeptide [15], whereas fungal FASN forms a 2.6-MDa barrel-shaped dodecameric complex that includes six α subunits (encoded by *FAS2*) and six β subunits (encoded by *FAS1*) [13]. The FASN-α subunit is known to be essential for lethal infections caused by *C. albicans*, *Candida parapsilosis*, and *Cryptococcus neoformans* in both rat oropharyngeal and systemic mouse infection models [16–19]. However, its pathogenic mechanism remains poorly understood, despite the suggestion of a fitness defect mechanism [17]. Here, we found that the essentiality of the FASN-α subunit in lethal *C. albicans* infection is primarily

mediated by the suppression of protective innate immune responses. Unlike the vast majority of mutants with defects in the bloodstream infection model [20], deleting the α subunit gene (*FAS2*) does not affect fungal growth, yeast-to-hypha morphogenesis, or intracellular fatty acid profiles when human serum is available. However, the deletion displays increased circulating innate immune responses and enhances activated neutrophil fungicidal activity through the unmasking of immunogenic cell wall epitopes, which facilitates fungal clearance and ultimately lowers fungal pathogenicity. Furthermore, priming mice with the *FAS2*-deleted mutant provided robust protection against subsequent lethal *C. albicans* infection, suggesting its potential as a tool for antifungal immunotherapeutic strategies.

## Results

### The avirulence of *C. albicans fas2Δ/Δ* is not primarily due to its growth defect *in vivo*

Consistent with previous studies [17], the survival rate of immunocompetent mice systemically challenged with the FASN-α subunit-encoding gene *FAS2* deletion mutant *fas2Δ/Δ* was 100%, and did not show any clinical signs of infection (S1 Fig). As immunocompromised individuals are extremely susceptible to invasive fungal infections, we next evaluated if this protective effect would also persist in the immunocompromised mouse model. Strikingly, as shown in Fig 1A, the difference in survival rates between the *fas2Δ/Δ*- and WT-infected cohorts was abolished in cyclophosphamide (CTX) pre-treated immunocompromised murine models administered intraperitoneally (i.p.), with all mice dying within 3 days, and both groups exhibited nearly identical abundances (~$10^5$ CFU/g kidney) in their kidneys (Fig 1B). The enhanced virulence of *fas2Δ/Δ* in immunodeficient mice indicates that an immunocompetent response is necessary for the avirulence of *fas2Δ/Δ in vivo*.

To explore whether the failure of the lethal infection caused by *fas2Δ/Δ* might be linked to a growth defect, as hypothesized by Zhao [17], we assessed its competitive fitness under various conditions. As expected, the auxotrophic phenotype of *fas2Δ/Δ* was severely defective in standard medium without the addition of fatty acids [17] (S2A–S2C Fig). However, this growth limitation was effectively compensated when *fas2Δ/Δ* was cultured in human serum (50%), which found to comprise ~80% of total fatty acids, including the unsaturated linoleic (C18:2n-6) and oleic (C18:1n-9) acids, as well as the saturated palmitic (C16:0) and stearic acids (C18:0) [21]. In human serum, *fas2Δ/Δ* exhibited normal competitive fitness compared to WT, with a similar lag phase, and the growth kinetics and the maximum growth rate achieved were slightly slower and lower than those of the WT strain (Fig 1C and 1D). To further examine the growth ability *in vivo*, tissue grinders and serum that mimic the host environment were used. The growth of *fas2Δ/Δ* was weaker than that of WT, but the CFU value of *fas2Δ/Δ* at 24 h was 50- to 150-fold higher than its initial CFU value (Fig 1E). In the spleen homogenates and serum, *fas2Δ/Δ* exhibited a 150-fold increase in CFU over 24 hours, reaching levels comparable to WT, indicating that *FAS2* deletion allows *C. albicans* to become capable of growing and persisting *in vivo*. We also assessed the effect of *FAS2* on the metabolic enzyme activity of FASN. Liquid chromatography/mass spectrometry (LC/MS) revealed that the intracellular fatty acid profile in *fas2Δ/Δ* was markedly disrupted compared to WT, with reduced levels of certain long-chain fatty acids, particularly unsaturated fatty acids. Additionally, FASN activity was partially affected. However, the addition of serum restored the fatty acid composition of *fas2Δ/Δ* to levels more similar to WT (Fig 1F–1I). These results imply that exogenous fatty acids in human serum can compensate for the fitness defects of *fas2Δ/Δ*.

Pathologic analysis of infected tissues provided further insight into the mutant's behavior *in vivo*. Histopathologic analysis of the kidneys showed significant tissue damage, dense

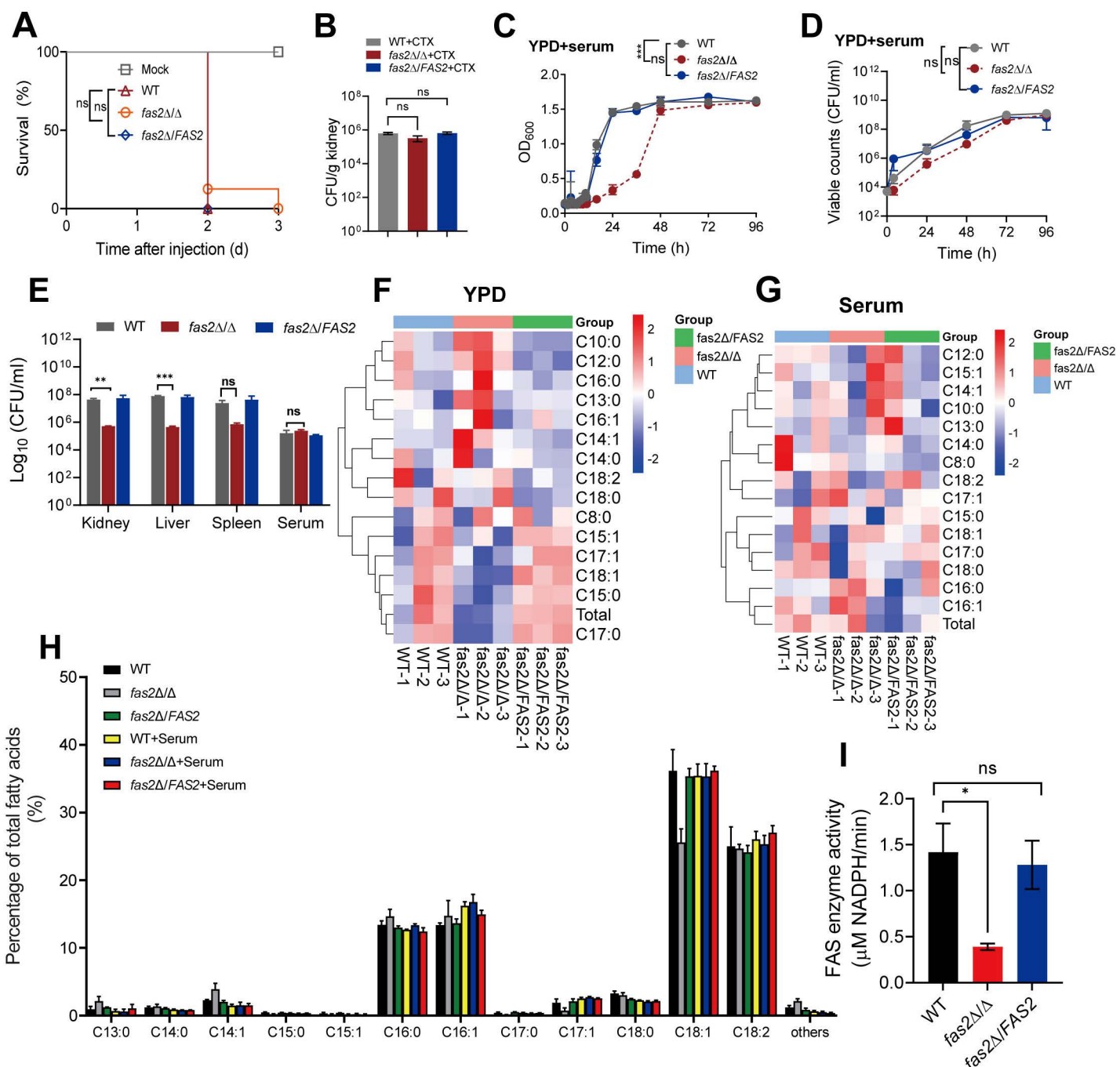

**Fig 1. Serum addition compensates for *fas2 Δ/Δ* growth defects.** (A) Immunosuppression was maintained *via* recurring intraperitoneal injections every 3 days with cyclophosphamide (CTX, 150 mg/kg of body weight), starting 4 days before infection (n = 4). Survival curves for CTX-immunosuppressed female BALB/c mice following intravenous injection with WT, *fas2Δ/Δ* and *fas2Δ/FAS2* (1 × 10⁴ CFUs per mice). (B) *C. albicans* fungal burden in kidneys of mice immunosuppressed with CTX 4 days postinfection (n = 4). (C, D) The growth of WT, *fas2Δ/Δ* and *fas2Δ/FAS2* in 50% human serum media over time determined by cell density (OD$_{600}$) (C) and viable cell counts (CFU/ml) (D). (E) *Ex vivo* growth assessment of the strains in fresh tissue homogenates of kidney, liver, and spleen (500 mg/ml) and 100% serum for 24 hours, measured by viable cell counts. (F-G) Fatty acid composition of *C. albicans* strains grown in YPD (F) and serum-containing media (G). Heatmaps display the relative abundance of different fatty acids, including saturated and unsaturated fatty acids, in WT, *fas2Δ/Δ*, and *fas2Δ/FAS2* strains. Each fatty acid is represented as a row, with color gradients indicating fold changes relative to WT. (H) Relative percentage of individual fatty acids in *C. albicans* strains grown in YPD and serum-containing media. The bar graph shows the distribution of specific fatty acids, including saturated and unsaturated types across WT, *fas2Δ/Δ*, *fas2Δ/FAS2*, and serum-supplemented groups. (I) The fatty acid synthase activity in WT, *fas2Δ/Δ* and *fas2Δ/FAS2*. Data are expressed as the mean ± SD of three independent experiments. Statistical significance is indicated by *$P < 0.05$, **$P < 0.01$, ***$P < 0.001$, with 'ns' for not significant. Two-way ANOVA was used for statistical analysis in C, D; the log-rank test was applied to survival data in A; and the two-tailed unpaired Student's t-test in B, E, and I.

hyphal formation, and substantial inflammatory cell infiltration 1 h after infection with WT (S1C Fig), but it was not observed in *fas2Δ/Δ*, indicating that *FAS2* is important for filamentation, a major virulence factor in *C. albicans* [20]. However, *fas2Δ/Δ* exhibited equivalent patterns of filamentation in the "germ tube" clinical diagnostic assay compared to WT, as well as in Spider and 10% FCS media at 37 °C, although there were more extensive pseudohyphae in the serum-containing medium (S2D and S2E Fig). Likewise, the RT-qPCR results showed the hyphae-associated genes were not significantly different between *fas2Δ/Δ* and WT (S2F Fig). Unlike yeast, hyphae are naturally invasive and express virulence factors such as adhesins, tissue-degrading enzymes, and the secreted toxin candidalysin [22]. The result of *fas2Δ/Δ* is dispensable for morphogenesis under *in vitro* conditions but not in the host, suggests that the reduced titers in deep organs (S1 Fig) were not substantially associated with major virulence factors of *C. albicans*.

## The protection conferred by *fas2Δ/Δ* in mice is host immune system dependent

Medzhitov *et al* have hypothesized that injury from infectious disease is the result of pathogen proliferation plus host intolerance of the pathogen [23]. To explore whether the elimination of lethal *C. albicans* infection induced by *fas2Δ/Δ* was mediated by the immune response, we measured five proinflammatory cytokines required for antifungal immunity in blood by ELISA [24]. As shown in Fig 2A, circulating levels of TNF-α, IL-1β, and IL-17a were elevated in mice infected with *fas2Δ/Δ* at 24 h post-infection compared to WT, whereas IFN-γ and IL-6 levels were significantly decreased but comparable to the saline (mock infection) control. By 8 days post-infection (d.p.i.), however, the elevated proinflammatory cytokines in *fas2Δ/Δ* mice had returned to baseline, limiting the potential for systemic chronic inflammation. In contrast, the production of all five cytokines in WT mice was massively increased throughout the time course, correlating with an ongoing cytokine storm, severe inflammation, and multiple organ failure during fungal sepsis. Monocytes, macrophages, and neutrophils are common sources of inflammatory cytokines during systemic *Candida* infections [25]. We next investigated whether the inflammatory response promotes the recruitment of immune cells in bloodstream. Using flow cytometry, monocytes (CD11b+Ly6Chigh), neutrophils (CD11b+Ly6G+), and macrophages (CD11c-CD11b+) were quantified in blood from 1 to 6 d.p.i. As shown in Fig 2B, the percentage and absolute number of monocytes (CD11b+Ly6Chigh), neutrophils (CD11b+Ly6G+), and macrophages (CD11c-CD11b+) significantly increased in *fas2Δ/Δ*-infected mice within 24 h in blood compared to WT, with neutrophil numbers being more pronounced, which is known to be critical in early fungal containment [26]. Notably, the cell populations of monocytes and macrophages in *fas2Δ/Δ*-infected mice started to decrease from 48 h post-infection, while neutrophils decreased from 72 h. All these immune cells returned to baseline by 6 d.p.i, similar to mock controls, contrasting with WT infections, where immune cells persisted throughout the observation period (Figs 2C–2E and S3A). These results indicate that *fas2Δ/Δ* triggers a strong but transient immune response after access to the bloodstream, but avoids prolonged immune activation. As fungal dissemination into the bloodstream is a critical step leading to invasive infections [27], we reasoned that the avirulence of *fas2Δ/Δ* is associated with enhanced immune defenses after access to the bloodstream, and thus decreased egress from the vasculature and penetration into target organs.

Kidney is a key site of *Candida* infection that leads to septic shock and mortality in mouse models of *Candida* infection [28]. As expected, *fas2Δ/Δ* induced significantly lower production of the proinflammatory cytokines TNF-α, IL-6, IL-1β, and IL-17a in whole kidney homogenates compared to WT at 1 d.p.i., albeit with increased levels compared to the mock controls, and all cytokines returned to baseline at 8 d.p.i (Fig 2F). We also assessed the impact

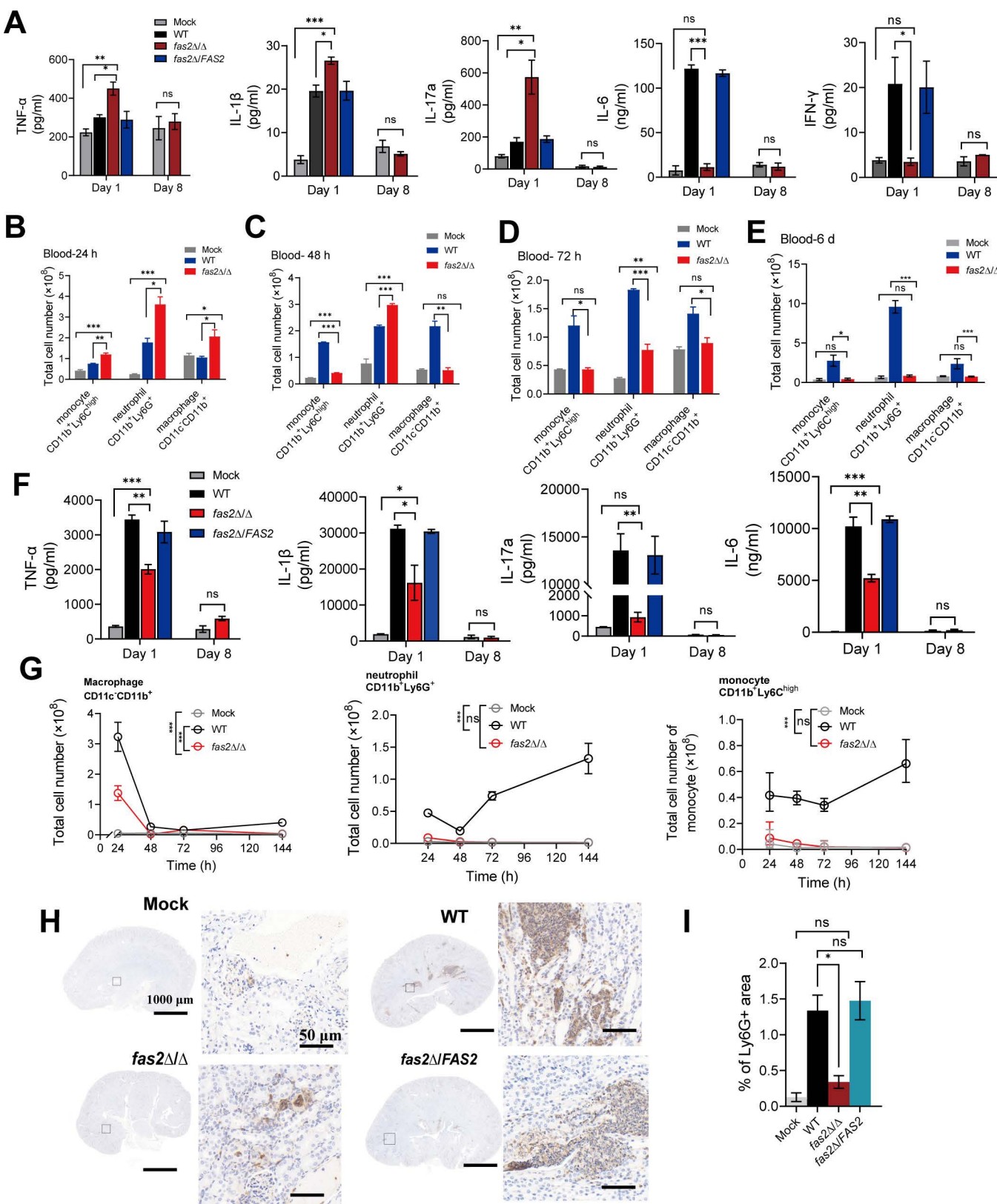

**Fig 2. Evaluation of the immune responses after infection with *fas2* Δ/Δ.** (A) Female BALB/c mice were infected with WT, *fas2*Δ/Δ, and *fas2*Δ/*FAS2* (5 × 10⁵ CFUs per mice), the serum levels of the proinflammatory cytokines TNF-α, IL-1β, IL-17a, IL-6, and IFN-γ concentrations at 1- and 8- days postinfection

were determined by ELISA (n = 3). (B-E) Peripheral blood of mice was analyzed at 24, 48, 72 h and 6 d after intravenous infection with *C. albicans* WT, *fas2Δ/Δ*, and *fas2Δ/FAS2* ($5 \times 10^5$ CFUs per mice) using FACS, with cells stained with antibodies to CD11b, CD11c, Ly6C and Ly6G, the absolute numbers of the different populations were statistically analyzed (n = 3). (F) Following intravenously injected with *C. albicans* WT, *fas2Δ/Δ*, and *fas2Δ/FAS2* ($5 \times 10^5$ CFUs per mice), the kidney homogenates of mice were analyzed at 1- and 8- days postinfection for the levels of TNF-α, IL-6, IL-1β, and IL-17a by ELISA (n = 3). (G) Kidney homogenates of mice were analyzed for immune cell populations at 24, 48, 72 h, and 6 d after intravenous infection with *C. albicans* WT, *fas2Δ/Δ*, and *fas2Δ/FAS2* ($5 \times 10^5$ CFUs per mice) using FACS stained with antibodies to CD11b, CD11c, Ly6C and Ly6G, the absolute numbers of the different populations were statistically analyzed (n = 3). (H, I) Representative images of kidney sections stained for the neutrophil marker Ly-6G (n = 3) (H) and quantification of kidney area scored as Ly-6G$^+$ (I) at 72h postinfection with WT, *fas2Δ/Δ* and *fas2Δ/FAS2* ($5 \times 10^5$ CFUs per mice) (n = 3). Data in (A, B, C, D, E, F, G, and I) are presented as mean ± SD. Statistical significance was assessed using a two-tailed unpaired Student's *t*-test for cytokine levels and blood immune cell populations (A, B, C, D, E, F, and I) or two-way ANOVA for kidney immune cell populations (G). Significance is indicated as *$P < 0.05$, **$P < 0.01$, ***$P < 0.001$, with ns denoting no significant difference.

of *FAS2* on infiltrating renal leucocytes (neutrophils, macrophages, and monocytes). During infection with WT, the relative amounts of renal neutrophils and monocytes remained elevated through the observation period of 144 h, whereas macrophages progressively decreased from 24 h onward (Fig 2G). By comparison, in *fas2Δ/Δ*-infected kidneys, although the percentage and absolute number of macrophages briefly rose at 24 h and returned to baseline at subsequent time points, the absolute numbers of neutrophils and monocytes remained comparable to mock controls throughout (Figs 2G and S3B). These patterns suggest that *fas2Δ/Δ* penetration from the vasculature to target organs attenuates inflammation-associated tissue damage. These results were consistent with renal histology Ly-6G staining (Fig 2H and 2I). The dUTP nick end label (TUNEL) staining and statistical analysis results revealed a markedly decreased number of dead cells in *fas2Δ/Δ*-infected mouse kidney tissues compared to WT, with only minimal increases over saline controls (S3C and S3D Fig), indicating that *fas2Δ/Δ* causes only minor, transient injury to the kidneys. At transcriptional level, RT-qPCR analysis of *KIM1* (kidney injury molecule-1), *NGAL* (neutrophil gelatinase-associated lipocalin), and *ICAM-1* (the major adhesion molecule), — markers of acute renal injury—revealed that during infection, *fas2Δ/Δ*-infected kidneys exhibited only a transient elevation of *ICAM-1* at 24 h, which was significantly lower than that of WT and returned to baseline at subsequent time points. In contrast, *KIM1* and *NGAL* levels were not different from those of the mock control throughout the time course (S3E Fig). Together, these results indicated that the avirulence of *fas2Δ/Δ in vivo* relies heavily on a rapid, self-limiting immune response.

## Neutrophils are the major cell type responsible for combating *C. albicans fas2Δ/Δ*

Due to the broad depletion of both the innate and adaptive arms of the immune system by CTX, it was unknown which primary immune cells were contributing to the clearance of *C. albicans fas2Δ/Δ in vivo*. Therefore, single-cell RNA sequencing (scRNA-seq) was applied to characterize the immunological profile of mice. scRNA-seq analysis of 113,892 single-cell transcriptomes (Fig 3A) revealed marked enrichment of neutrophil lineage genes (*ELANE* and *LTF*) as well as classical and non-classical monocytes in the spleens of *fas2Δ/Δ*-infected mice compared to WT (S4 Fig and S1–S3 Data files.) Nonetheless, the profiles of adaptive immune cells in mice, including B cells and T lymphocytes, exhibited no significant alterations between groups. These findings suggest that the innate antifungal immune responses, particularly those mediated by neutrophils and monocytes, might play an essential role in controlling *fas2Δ/Δ*-mediated *C. albicans* infection.

Neutrophils and macrophages play a critical role in antifungal innate immunity and function as the first line of defense against fungal infections [25]. To investigate whether neutrophils contribute to host protection against *C. albicans fas2Δ/Δ*, neutrophils were depleted by

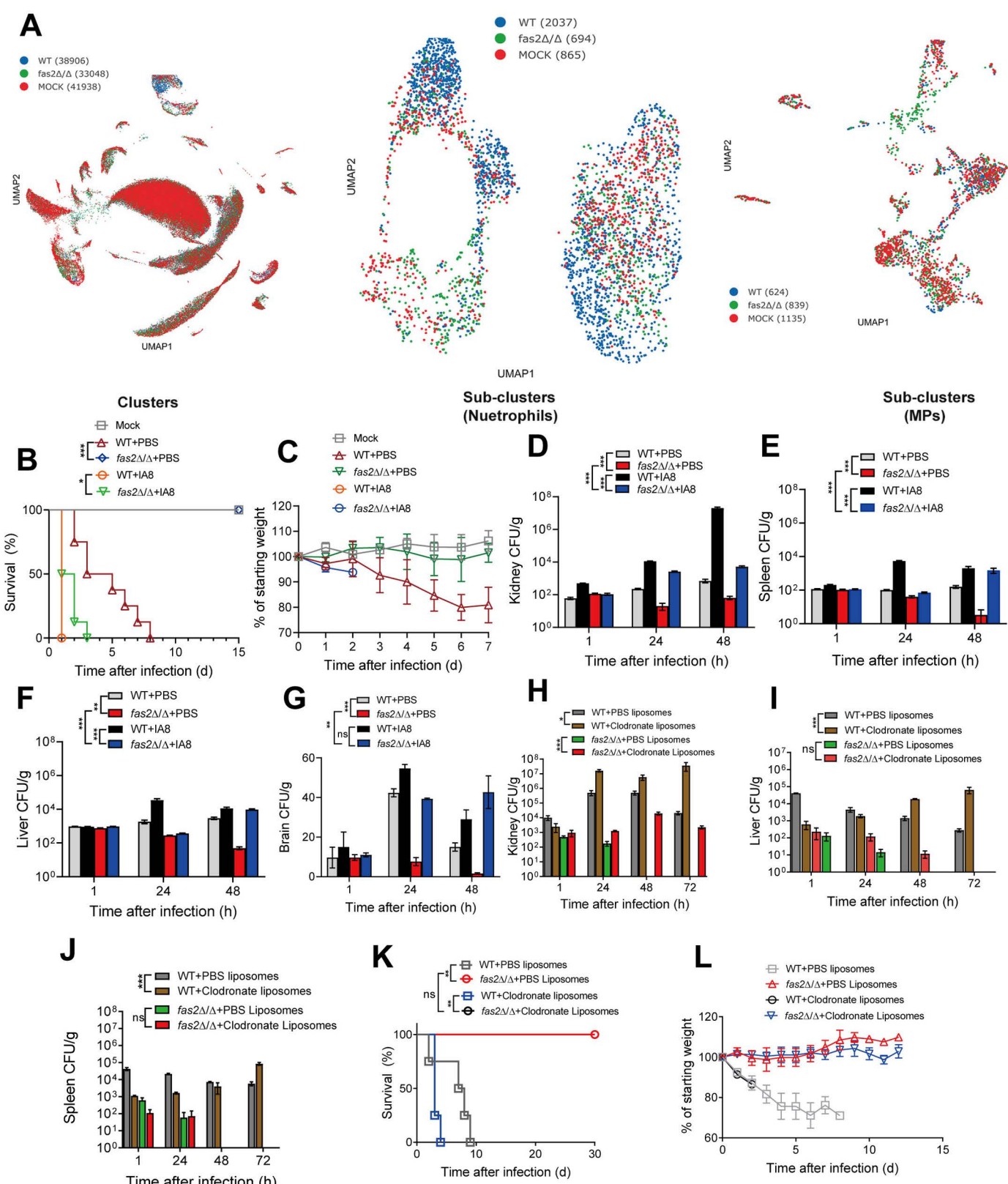

**Fig 3. Neutrophils are required for the *fas2* Δ/Δ mutant induced protective immunity.** (A) UMAP plots showing the distribution of immune cell clusters in WT, *fas2*Δ/Δ, and mock-infected mice. The left panel displays overall immune cell clustering by sample group, while the middle and right panels highlight

neutrophil and monocyte phagocyte (MP) sub-clusters, respectively. The number of cells analyzed from each group is indicated in the legend. (B, C) Mice treated with Ly6G (IA8) for neutrophil depletion by recurring intraperitoneal injections of 300 μg of anti-mouse Ly6G (IA8) antibody every 2 days, beginning one day before infection. At day 0, mice were intravenously injected with WT and *fas2Δ/Δ* (2 × 10⁵ CFUs per mice), survival rates (B) (n = 8) and weight loss (C) (n = 8) of mice post infection are shown. (D-G) Fungal burdens in kidneys (D), spleens (E), livers (F), and brains (G) of mice at 1, 24, 48 h post infection with WT and *fas2Δ/Δ* (2 × 10⁵ CFUs per mice) with PBS drug vehicle (n = 3) or Ly6G (IA8) (n = 3) treatment. (H-J) Fungal burdens in kidneys (H), livers (I), and spleens (J) of mice administered PBS or clodronate liposomes 24 h before and 24 h after intravenous infection with WT and *fas2Δ/Δ* (2 × 10⁵ CFUs per mice) (n = 3). (K, L) Mice treated with clodronate liposomes for macrophage depletion. Survival curves (K) and weight loss (L) in mice given PBS liposomes or clodronate liposomes 24 h before and 24 h after intravenous infection with WT and *fas2Δ/Δ* (2 × 10⁵ CFUs per mice) (n = 4). Data from three independent experiments are present as the mean ± SD. Statistical significance was determined using two-way ANOVA for fungal burdens (D, E, F, G, H, I, and J) or log-rank test for survival curves (B, K). Significance is indicated as *$P < 0.05$, **$P < 0.01$, ***$P < 0.001$, with ns denoting no significant difference.

administering 200-μg intraperitoneal injections of the IA8 antibody, 24 h prior to *C. albicans* infection and every 48 h thereafter to female BALB/c mice (6–8 weeks old) [29]. As shown in Fig 3B and 3C, all mice succumbed to death within 5 days after infection with *fas2Δ/Δ* due to acute neutrophil depletion, which showed no statistically significant difference compared to WT. Likewise, etiologic analysis results showed that upon IA8 depletion, a persistence of fungal burdens in the kidneys, livers, spleens, and brains was observed in *fas2Δ/Δ*-infected mice. At 2 d.p.i., mice infected with WT exhibited an average of 6.63×10⁵ CFU/g kidney, while mice infected with *fas2Δ/Δ* showed an average fungal burden of 1.08×10⁵ CFU/g kidney, which represented only an ~6-fold difference in fungal burden between the two groups (Fig 3D–3G). These findings highlight the critical role of neutrophils in clearing *fas2Δ/Δ* infections. To assess the role of macrophages, a similar strategy was employed in macrophage-normal and macrophage-depleted mice using PBS or clodronate liposomes, respectively [30]. As shown in Fig 3H, the depletion of macrophages (clodronate liposome) led to a persistent fungal burden in the kidneys of *fas2Δ/Δ*-infected mice, with the CFU value reaching approximately 1/2 of that observed in WT mice receiving liposomes at 3 d.p.i. However, *fas2Δ/Δ* was cleared from the livers (Fig 3I) and spleens (Fig 3J). The abundance of the fungus in these organs, as well as the survival rates (Fig 3K) and weight loss (Fig 3L), did not significantly differ between macrophage-depleted and normal mice infected with *fas2Δ/Δ*. In contrast, *C. albicans* WT stably persisted in the infected organs of the macrophage-depleted mice (clodronate liposome), and the fungal burdens were much higher than those in the normal mice (PBS liposome). These results indicate that while macrophages may play a secondary role in controlling *fas2Δ/Δ* infection. In contrast neutrophils remain the primary immune cell type responsible for clearing this strain *in vivo*.

## The *fas2Δ/Δ* mutant triggered the phagocytosis and fungal-killing activities of neutrophils

To test the hypothesis that neutrophils are directly activated by exposure to *C. albicans fas2Δ/Δ*, we performed *in vitro* co-culture experiments with bone-marrow (BM)-derived neutrophils. Unlike WT *C. albicans* strain, *fas2Δ/Δ* trigger a strong IL-1β and TNF-α secretion into culture supernatants, levels of other cytokines were below the limits of detection for the assay (Fig 4A and 4B). To determine whether direct cell-cell contact is required for neutrophils activation by *fas2Δ/Δ*, we repeated the experiment under two conditions: one allowing direct co-culture of two cells in the same well, and the other separating two cells by a porous membrane to permit only secreted factors to pass. Using RT-qPCR to measure the transcript levels of pro-inflammatory cytokine, we found that *fas2Δ/Δ* triggered IL-1β and TNF-α responses from neutrophils only when the cells were in direct physical contact (Fig 4C and 4D), suggesting that cell-cell contact is required for *fas2Δ/Δ*-mediated neutrophils activation. We next examined whether cell viability of *fas2Δ/Δ* is necessary for their ability to suppress activation

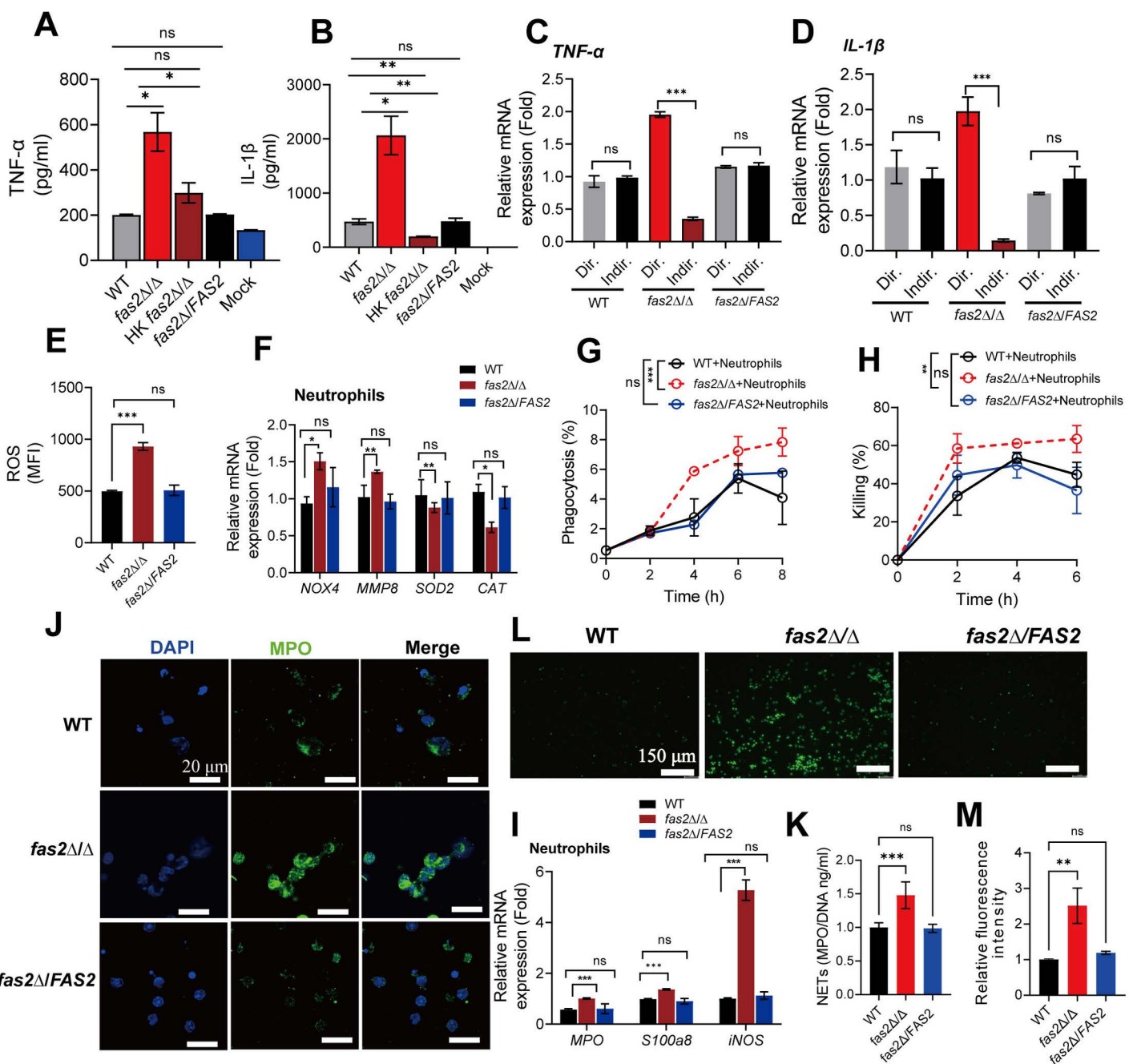

**Fig 4. The *fas2* Δ/Δ mutant promotes cytokine production and antifungal activities of neutrophils.** (A, B) BM-derived neutrophils from BALB/c mice were cultured with WT, *fas2Δ/Δ*, *fas2Δ/FAS2*, and heat-killed (HK) forms of *fas2Δ/Δ* at a MOI = 1 for 2 h followed by measurement of the proinflammatory cytokines levels of TNF-α (A) and IL-1β (B) in cell supernatants by ELISA (n = 3). (C, D) BM-derived neutrophils from BALB/c mice were cultured in the same well with WT, *fas2Δ/Δ*, or *fas2Δ/FAS2* (Dir) or separately using a transwell system (Indir) at a MOI of 1 for 2 h, followed by assessment of *TNF-α* (C) and *IL-1β* mRNA by RT-qPCR (D) (n = 3). (E) BM-derived neutrophils were stimulated with WT, *fas2Δ/Δ* and *fas2Δ/FAS2* (n = 3) for 2 h, followed by measurement of ROS production via a fluorescent ROS probe and the mean fluorescence intense (MFI) analysis (n = 3). (F) BM-derived neutrophils were co-cultured with WT, *fas2Δ/Δ*, and *fas2Δ/FAS2*, followed by RT-qPCR analysis of ROS synthesis-related genes expression (n = 3). (G, H) Phagocytosis (G) and killing capacity of neutrophils (H) after infecting WT, *fas2Δ/Δ*, and *fas2Δ/FAS2* at a MOI = 1 (n = 3). (I) BM-derived neutrophils were co-cultured with WT, *fas2Δ/Δ*, and *fas2Δ/FAS2*, followed by RT-qPCR analysis of neutrophil-derived protein expression (n = 3). (J) Representative fluorescence images of BM-derived neutrophils of mice infected with WT, *fas2Δ/Δ*, and *fas2Δ/FAS2*, stained with antibodies against MPO (green) and nuclei (DAPI, blue), with a merged image shown. (K) Quantitative analysis of MPO fluorescence intensity in BM-derived neutrophils (n = 4). (L, M) Neutrophils were incubated with WT, *fas2Δ/Δ*, and *fas2Δ/FAS2* (MOI = 2:1), Sytox Green was added after 180 min of incubation to detect extracellular DNA. A representative image of NETs induced by each strain is shown (L). Quantifying of relative Sytox Green fluorescence intensity in each group from

three independent experiments was performed (M). Data from three independent experiments are presented in (A, B, C, D, E, F, G, H, I, K, and M) as the mean ± SD. Statistical analysis was performed using two-way ANOVA for phagocytosis and killing assays (G, H) or two-tailed unpaired Student's t-test for ROS measurements and gene expression analyses (A, B, C, D, E, F, H, I, K, and M). Significance is indicated as *$P < 0.05$, **$P < 0.01$, ***$P < 0.001$, with ns denoting no significant difference.

of neutrophils. As shown in Fig 4A and 4B, co-incubation of neutrophils with live but not heat-killed (HK) *fas2Δ/Δ* resulted in the secretion of IL-1β and TNF-α. Together, these results suggest that *fas2Δ/Δ* can activate inflammatory response by neutrophils, but live *C. albicans* WT are able to block immune activation in a *FAS2*-dependent fashion.

Neutrophils and macrophages are professional phagocytes of the innate immune system that are essential in controlling fungal infection by phagocytosis and killing mechanisms [2]. Next, we investigated whether the inflammatory phenotype of BM-derived neutrophils with *fas2Δ/Δ* was related to its antifungal activity, including fungal-killing capacity *ex vivo*, phagocytosis, neutrophil extracellular traps (NETs), or reactive oxygen species (ROS) [31]. ROS plays an important role in the initial step of fungal killing in phagosomes [32]. As shown in Fig 4E, exposure to *fas2Δ/Δ* but no WT or *fas2Δ/FAS2* induced a strong ROS production in coculture assays with BM-isolated neutrophils. Likewise, co-incubation of neutrophils with *fas2Δ/Δ* resulted in significant upregulated genes expression associated with ROS synthesis (*NOX4*, *MMP8*), whereas the genes associated with ROS clearance (*CAT*, *SOD2*) were significantly downregulated (Fig 4F). Additionally, compared to WT *C. albicans*, significantly increased phagocytosis (Fig 4G) and fungicidal activity (Fig 4H) of neutrophils were well associated with the increased ROS production induced by *fas2Δ/Δ*. These results indicate that at least part of the fungicidal effect of neutrophil against *fas2Δ/Δ* depends on ROS production. NETs are a natural defense barrier composed of MPO, neutrophil elastase and histones, and NETs are formed by neutrophils after stimulation with pathogenic infections [33]. Using RT-qPCR to quantify proinflammatory MPO, S100a8, as well as iNOS expression in neutrophils, we observed significant upregulation after 6 h of infection with *fas2Δ/Δ* compared to WT (Fig 4I). In support of neutrophils activation by *fas2Δ/Δ*, immunofluorescent staining revealed *fas2Δ/Δ* triggered stronger degranulation of BM-derived neutrophils, as reflected by elevated infiltration of MPO (Fig 4J and 4K). Additionally, Sytox green staining revealed a higher DNA content within NETs of *fas2Δ/Δ*-stimulated neutrophils, thereby substantiating the enhanced formation of NETs in response to the *fas2Δ/Δ* stimulus (Fig 4L and 4M). The experiments demonstrate that the increased phagocytic and killing potential of neutrophils is critical for the elimination of *C. albicans fas2Δ/Δ* during systemic infection.

Compatible with the hypothesis that there was a limited role for *fas2Δ/Δ* in triggering the antifungal response by macrophage activation, we observed no significant difference in phagocytic efficiency and the fungal-killing capacity of BM-derived macrophages (BMDMs) between the WT and *fas2Δ/Δ*-stimulated groups (S5A and S5B Fig). These results were consistent with the comparable levels of pro-inflammatory cytokine production upon WT- and *fas2Δ/Δ*- stimulation with BMDMs (S5C–S5E Fig). Additionally, *fas2Δ/Δ* resulted in similar MPO, S100a8, and iNOS transcript levels compared to WT (S5F Fig), as well as ROS activity during co-incubation with BMDCs (S5G Fig), indicating that the elimination of *fas2Δ/Δ* by the immune response was not likely dependent on macrophage alone. These results were further confirmed by co-culture experiments using RAW264.7. After phagocytosis, the WT *C. albicans* control strain formed elongated hyphae and typically lysed the macrophages by 2h. In contrast, the majority of *fas2Δ/Δ* either formed small hypha-like outgrowths or failed to undergo detectable growth within the macrophage (S5H Fig); however, after 12 h, the mutant formed hyphae and escaped from macrophages, resulting in increased *C. albicans* survival rates and macrophage cell damage (S5I–S5K Fig). Moreover, the production of intracellular

NO and *iNOS* expression in RAW264.7 macrophages infected with *fas2Δ/Δ* was no different from that of WT (S5L and S5M Fig). Taken together, our data indicate that *fas2Δ/Δ* primarily activates neutrophils, boosting their antifungal response while bypassing similar activation in macrophages.

## The immunomodulatory role of *fas2Δ/Δ* is mediated by the exposure of immunogenic cell wall epitopes

Global lipidomics and untargeted metabolomics analyses were applied to profile the metabolic changes associated with *FAS2* deficiency in *C. albicans*. The lipidomic analysis identified 20 significant differential lipid compounds out of 146 (*P* < 0.05, VIP ≥1) between *fas2Δ/Δ* and WT strains (Fig 5A and 5B and S4 Data). Differentially enriched metabolites were assigned to biochemical pathways based on KEGG (http://genome.jp/kegg/), showing that glycosyl-phosphatidylinositol (GPI)-anchor biosynthesis and glycerophospholipid metabolism were the most affected pathways in *fas2Δ/ΔΔ* (Fig 5C and S5 Data). Interestingly, these results were consistent with the metabolomics analysis, which indicated that among the significant differential metabolites between WT and *fas2Δ/Δ*, lipids and lipid-like molecules (32%) were the primary superclass (Fig 5D and 5E). Specifically, glycerophospholipids and fatty acyls, such as LPCs, PIs, PC, and LPEs accounted for the majority (S6 and S7 Data files). The modifications of glycerophospholipid metabolism and GPI-anchor biosynthesis play vital roles in fungal cell wall structure and cell surface recognition by the innate immune system [34], implying an alteration in the cell wall components in *fas2Δ/Δ*. The multi-omics data, combined with transmission electron microscopy (TEM) observations of a significant thicker cell wall in *fas2Δ/Δ* (0.203 ± 0.005 μm) compared to WT (0.157 ± 0.005 μm, *P* < 0.05) (Fig 5F), indicating alterations in the cell wall of *fas2Δ/Δ*.

The fungal cell wall consists of an inner layer enriched in β (1,3)-glucan and underlying chitin, and an outer layer of mannoproteins proteins [35]. No difference was observed in outer mannan structures between WT and *fas2Δ/Δ*, suggesting that *fas2Δ/Δ* undergoes changes in inner cell wall structuring (Fig 5F). Accordingly, we found that *fas2Δ/Δ* was much more hypersensitive to the inner layer of the cell wall stressors, including the chitin-binding agents calcofluor white (CFW), Congo Red (CR), as well as the β (1,3)-glucan synthase inhibitor caspofungin (CAS) (S6A Fig). Neutrophils have been shown to be capable of recognizing exposed β (1,3)-glucan to facilitate in fungal clearance [36]. As shown in S6B Fig, *fas2Δ/Δ* was much more sensitive to β-glucanase compared to WT, which serves an indirect measure of β (1,3)-glucan exposure [37], indicating that β-glucan is more accessible in *fas2Δ/Δ*. Consistently, fluorescently labeled antibodies binding revealed increased staining for β-glucan on *fas2Δ/Δ* (Fig 5G). Flow cytometry further confirmed a nearly 4-fold increase in surface-exposed β-glucan in *fas2Δ/Δ* compared to WT, as shown by flow cytometry (Fig 5H and 5I). However, the increased in β-glucan exposure was greatly reversed in *fas2Δ/FAS2*. Additionally, the total β (1,3)-glucan contents, as assessed by aniline blue, were dramatically increased in *fas2Δ/Δ* (S6C and S6D Fig), which was consistent with the transcriptional levels of β (1,3)-glucan synthase genes *FKS1/GSC1* that were upregulated in *fas2Δ/Δ* (Fig 5J), suggesting that the loss of *FAS2* induces the unmasking of β (1,3)-glucan in *C. albicans*.

To investigate alterations in other immunogenic cell wall epitopes, we probed the *C. albicans fas2Δ/Δ* with CFW, wheat germ agglutinin (WGA) and concanavalin A (ConA) to assess total and exposed levels of chitin, as well as mannan, respectively. The flow cytometry results and fluorescence microscopic images exhibited a significant increase in both total (Fig 5K and 5L) and exposed (Fig 5M and 5N) chitin in *fas2Δ/Δ* compared to WT. Also, microscopy confirmed a significant de-cloaking of chitin around the cell periphery in *fas2Δ/Δ*, with intense WGA staining (Fig 5G). Nevertheless, there was no perceptible difference in ConA

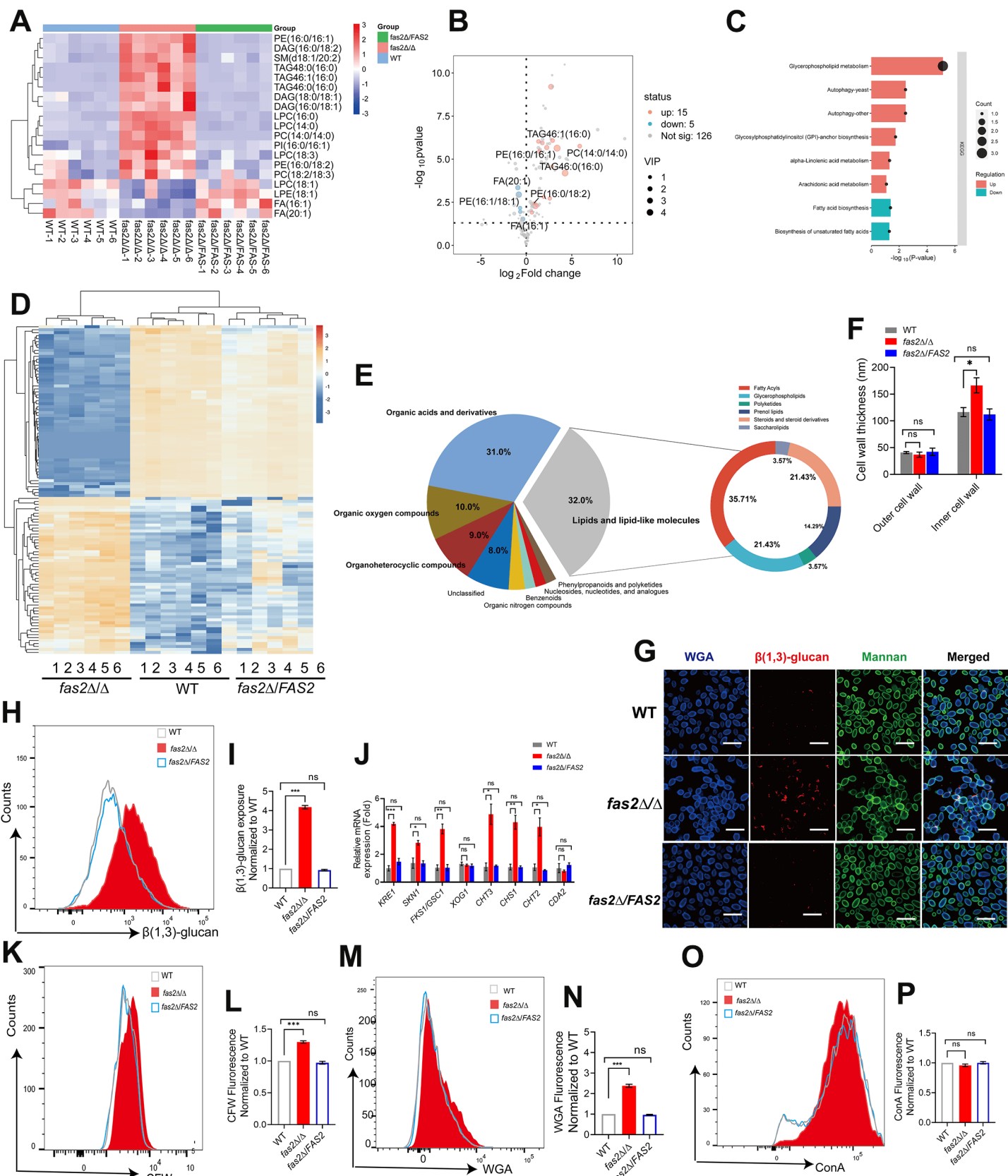

**Fig 5. The *fas2Δ/Δ* mutant induces the immunogenic epitopes exposure of the fungal cell wall.** (A) Heatmap depicting the relative abundance of differential metabolites in WT, *fas2Δ/Δ*, and *fas2Δ/FAS2*. Color intensity transitions from blue (low) to orange (high) indicate metabolite levels. (B) Volcano plot describing the

abundance difference of metabolites between WT and *fas2Δ/Δ*. The vertical dashed lines on each side correspond to −1.0 and 1.0 cut points. Log2Fold Change thresholds, while the horizontal dashed line represents the *P* value cutoff of 0.05 for differential metabolites identification. (C) Enriched metabolic pathways (*P* < 0.05) were identified among the top 20 differentially abundant metabolites in WT versus *fas2Δ/Δ*. (D) Heatmap showing differential metabolites abundance in WT, *fas2Δ/Δ*, and *fas2Δ/FAS2* by using untargeted metabolomics analysis. (E) The pie chart shows the class of the top 30 differential metabolites in WT and *fas2Δ/Δ* screed by untargeted metabolomics. (F) Electron micrographs highlighting the ultrastructure of the cell wall of WT, *fas2Δ/Δ*, and *fas2Δ/FAS2*, and quantification of the thickness of the inner cell wall layer and mannoprotein fibril length. Data points represent the mean ± SEM from five independent cells, each measured at 30 distinct cell periphery points. (G) Mid-log phase cultures of WT, *fas2Δ/Δ*, and *fas2Δ/FAS2* were stained with anti-β (1,3)-glucan antibody, fluorescein conjugated wheat germ agglutinin (WGA) and Alexa Fluor 488 conjugated concanavalin A (ConA) to assess glucan, chitin, and mannan content, respectively. Representative confocal microscopy images are shown. (H, I) Overnight cultures of WT, *fas2Δ/Δ*, and *fas2Δ/FAS2* strains were stained with anti-β (1,3)-glucan antibody to assess glucan. Representative histograms (H) and MFI normalized to WT (I) from three independent experiments alongside standard deviation are shown. (J) Relative mRNA expression levels of the genes involved in the cell wall synthesis in WT, *fas2Δ/Δ* and *fas2Δ/FAS2*. (K–P) Overnight cultures of WT, *fas2Δ/Δ*, and *fas2Δ/FAS2* strains were stained with anti-β (1,3)-glucan antibody, CFW, fluorescein conjugated wheat germ agglutinin (WGA) and ConA to assess glucan, total chitin, surface exposed chitin, and mannan, respectively. Representative histograms (K, M, and O) and MFI normalized to WT (L, N, and P) from three independent experiments alongside standard deviation are shown. Statistical significance was determined using a two-tailed unpaired Student's t-test. Significance is indicated as *$P$ < 0.05, **$P$ < 0.01, ***$P$ < 0.001, with ns denoting no significant difference.

binding between the two groups, further confirming that *fas2Δ/Δ* did not induce changes in the outer cell wall (Fig 5O and 5P). To further explore how *fas2Δ/Δ* affects cell wall carbohydrate homeostasis, we measured the transcript levels of the major cell wall synthesis genes, including *KRE1* and *SKN1* (β (1,6)-glucan), *FKS1/GSC1* (β (1,3)-glucan) and *XOG1* (α-1,3-glucan). In *fas2Δ/Δ*, the transcript levels of *KRE1, SKN1* and *FKS1/GSC1* were significantly upregulated, whereas the expression of *XOG1* was not statistically altered (Fig 5J). In addition to genes involved in glucan synthesis, the RT-qPCR results for the chitin synthesis genes in *fas2Δ/Δ* showed that *CHT3* (~3 fold), *CHS1* (~4 fold) and *CHT2* (~4 fold) were upregulated, while the chitin/chitosan deacetylase gene *CDA2* was downregulated (Fig 5J). Taken together, our results indicate that the loss of *FAS2* enhances the exposure of immunogenic β (1,3)-glucan and chitin epitopes on the cell wall.

## *FAS2* impacts *C. albicans* cell wall properties *via* the Rho-1 dependent Mkc1-MAPK pathway

The process of cell wall biogenesis and remodeling is regulated by several complex signaling pathways, such as mitogen-activated protein kinase (MAPK) cascades (including Cek1 and Mkc1-dependent), Hog pathway, the calcineurin pathway, the PKA/cAMP pathway, and the Rim/alkaline response pathways [38]. Among these MAPK pathways, the Mkc1-MAPK cascade, which consists of Bck1-Mkk2-Mkc1, is primarily involved in cell wall biogenesis and responses to external cell wall stress, oxidative stimuli, and antifungal drugs [39]. As shown in Fig 6A, compared with WT or *fas2Δ/FAS2*, the transcriptional levels of the upstream proteins Rho1, Pkc1, Bck1, and Mkk2 in the canonial Mkc1-MAPK cascade, as well as the transcription factors Swi4/Swi6 and Rlm1 in the Mkc1-MAPK pathway were consistently upregulated in *fas2Δ/Δ* (Fig 6B). In addition, phosphorylation of Mkc1 was increased in *fas2Δ/Δ* compared with WT or *fas2Δ/FAS2*, indicating that Mkc1-MAPK cascade is activated by *FAS2* disruption (Fig 6C and 6D). This further confirmed the disruption of cell wall integrity in *fas2Δ/Δ* and this strain's hypersusceptibility to cell wall stressors (S6A Fig). Rho1, an upstream protein, transmits the signal toward the Mkc1-MAPK cascades and is also a well-known regulatory subunit of β (1,3)-glucan synthase [40,41], strongly increased GTP-Rho1 activity in *fas2Δ/Δ* with respect to that of WT (Fig 6E). Moreover, the phosphorylation of Rho1-activated protein Pkc1, a necessary factor for regulating the downstream transcription factors of the Mkc1-MAPK cascade [39], was significantly increased in *fas2Δ/Δ* compared to WT (Fig 6F and 6G). Furthermore, treatment with staurosporine (STS), an exogenous inhibitor of Pkc1, led to significantly decreased Mkc1 phosphorylation (Fig 6D) and reduced β (1,3)-glucan content in *fas2Δ/Δ* compared to WT (Fig 6H and 6I). Intriguingly, the addition of exogenous STS

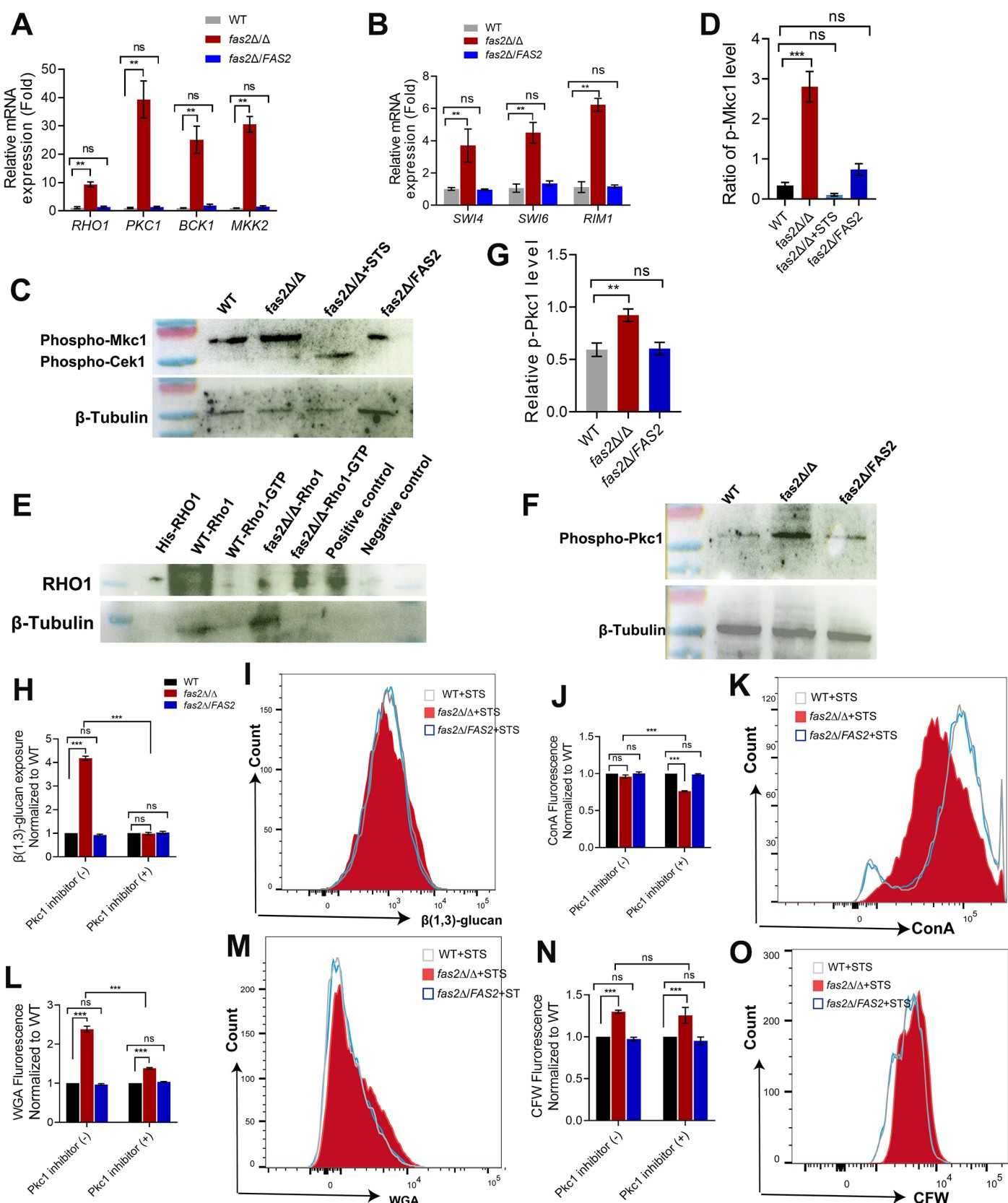

**Fig 6. The *FAS2* deletion induces β-glucan unmasking through Rho-1 dependent Mkc1-MAPK pathway.** (A) Relative mRNA expression levels of the genes involved in the Mkc1-MAPK pathway in WT, *fas2Δ/Δ* and *fas2Δ/FAS2* were assessed by RT-qPCR. (B) Relative mRNA expression levels of the Mkc1-MAPK

pathway transcription factors Swi4/Swi6 and Rim1 in WT, *fas2Δ/Δ* and *fas2Δ/FAS2* were assessed. (C) Protein extracts from WT, *fas2Δ/Δ* and *fas2Δ/FAS2* at the mid-log phase were analyzed for western blot using an anti-P44/42 antibody to detect phosphorylated Mkc1and Cek1. (D) Quantification of phosphorylated Mkc1 to the β-Tubulin fraction in WT, *fas2Δ/Δ* and *fas2Δ/FAS2*, as well as *fas2Δ/Δ* in the presence (+) of STS, represented as Fold-change (n = 3 biological replicates across 3 separate blots). (E) Activation of GTP-bound form of Rho1 in WT, *fas2Δ/Δ* and *fas2Δ/FAS2* were evaluated using the Rho activation assay biochem kit, with β-Tubulin as a loading control. (F, G) Pkc1 phosphorylation levels were determined by western blot using an anti-phosphoserine PKC substrate antibody in WT, *fas2Δ/Δ* and *fas2Δ/FAS2* (F), and quantification of band intensities relative to β-Tubulin is shown (G). (H-N) WT, *fas2Δ/Δ* and *fas2Δ/FAS2* strains were cultured in the absence (-) or presence (+) of STS, and cell wall components were assessed by staining with anti-β (1,3)-glucan antibody (H, I), ConA (J, K) for mannan, WGA for surface-exposed chitin (L, M) and CFW (N, O) for total chitin. The MFI from three independent experiments is displayed with standard deviation. Data in (A, B, D, G, H, J, L, and N) are presented as mean ± SD. Statistical significance was determined using a two-tailed unpaired Student's t-test, indicated as $*P < 0.05$, $**P < 0.01$, $***P < 0.001$, with ns denoting no significant difference.

did not significantly decrease the content of mannan (Fig 6J and 6K) and chitin (Fig 6L–6O) in *fas2Δ/Δ* as shown by flow cytometry. Therefore, these results suggest that *FAS2* deletion results in cell wall remodeling in *C. albicans* mainly *via* the Rho1-dependent Mkc1-MAPK pathway. The Cek1-MAPK cascade has been suggested to control β (1,3)-glucan masking, and activation of this pathway increases β-glucan exposure in *C. albicans* [40]. However, phosphorylation of Cek1 was no perceptible difference in *fas2Δ/Δ* compared with WT (Fig 6C), indicating that the disruption of *FAS2* does not activate the Cek1-MAPK pathway.

## Priming with *fas2Δ/Δ* protects mice from subsequent lethal *C. albicans* infection

The above results suggest that the avirulence of *C. albicans fas2Δ/Δ* might be due to the enhanced innate immune responses mediated by exposed β-glucans in the cell wall skeleton, a well-known inducer of trained immunity [42]. Thus, we questioned whether priming with *fas2Δ/Δ* could protect the host from subsequent lethal *C. albicans* infection. Female BALB/c mice (6–8 weeks old) were intravenously primed with *fas2Δ/Δ* and subsequently challenged with a fully virulent WT *C. albicans* strain at various time points (3, 7, 15, 28, and 60 days) (Fig 7A). All mice were infected with each *C. albicans* strain using a lethal dose of $5 \times 10^5$ CFUs and monitored daily for survival. Strikingly, when the interval of inoculation was 7 days, ≥ 87.5% of mice primed with *fas2Δ/Δ* survived for more than 25 days (Fig 7B). Consistently, kidney damage and fungal loads were significantly reduced in these mice after reinfection with WT (Fig 7C and 7D). Protection remained effective at 3 months, with over 25% of mice surviving the lethal challenge, suggesting that the protective effect could last at least until this time point. Thus, our results showed that priming with *fas2Δ/Δ* effectively protects mice from lethal *C. albicans* challenge, even though the immune protection was not consistently durable. When the interval of inoculation was increased to 15 days, a slight decrease in protective efficacy was observed, with ~50% of mice primed with *fas2Δ/Δ* surviving for over 25 days (Fig 7E). In contrast, although a statistically significant protective effect was observed when the interval of inoculation was decreased to 3 days (Fig 7F) or increased to 28 days (Fig 7G), all animals eventually succumbed to death. Moreover, when the interval of inoculation was increased to 60 days (Fig 7H), priming with *fas2Δ/Δ* provided no protection against *C. albicans* reinfection.

To determine whether the protection of *fas2Δ/Δ* is controlled by the FASN-α subunit, we constructed the complemented strain *fas2Δ/FAS2* by reintroducing a *FAS2* gene ORF into the *FAS2* locus of *fas2Δ/Δ*, and mice were primarily challenged with the *fas2Δ/FAS2* strain. However, the *FAS2* revertant strain ($5 \times 10^5$ CFUs) did not protect against *C. albicans* reinfection, and all animals died within 5 days (Fig 7B). In addition, the *fas2Δ/Δ* mutant inactivated by heat shock showed only a marginal protective effect. Together, these data suggest that the live *fas2Δ/Δ* mutant provided specific, albeit temporary, protection against lethal *C. albicans* infection, regulated by FASN-α subunit.

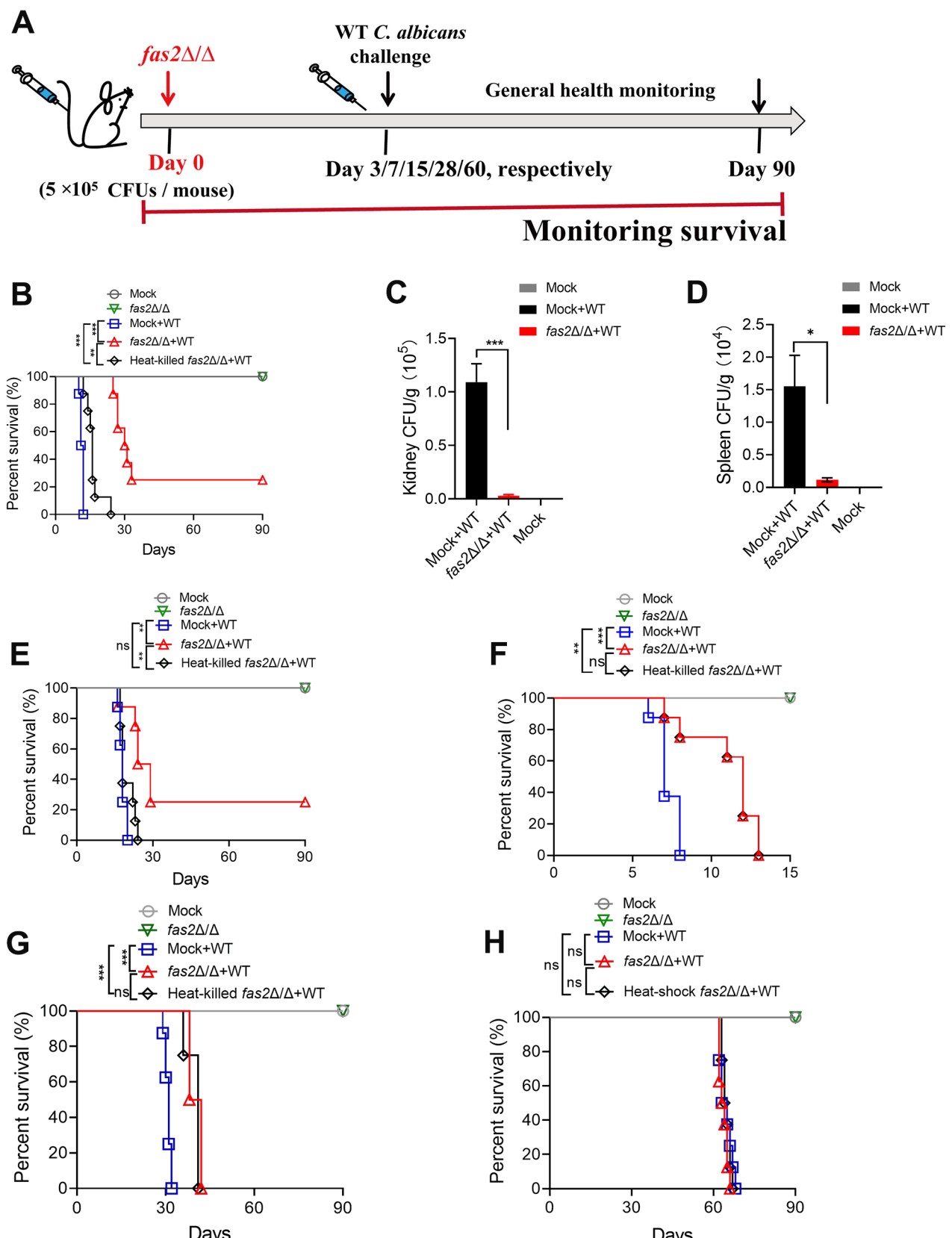

**Fig 7. Mice systemically primed with *fas2* Δ/Δ mutant are significantly protected from systemic candidiasis.** (A) Schematic of the experimental design. Female BALB/c mice (6–8 weeks old) were intravenously immunized with $5 \times 10^5$ CFUs of the *fas2*Δ/Δ mutant and subsequently challenged

with a lethal dose of a fully virulent WT *C. albicans* strain at intervals of 3, 7, 15, 28, and 60 **d** (n = 8), and the survival rate was evaluated based on the number of days from primary infection with *fas2Δ/Δ* to reinfection with WT *C. albicans*. (B) Survival curves for mice systemically challenged with $5 \times 10^5$ CFUs of WT *C. albicans* after being infected with live, heat-killed *fas2Δ/Δ* mutant, or mock-vaccinated (normal saline control) for durations of 7 days (n = 8). (C, D) Kidney (C) and Spleen (D) fungal burdens of mice (n = 3) at 3 days postchallenge were determined by quantitative analysis. (E-H) Survival curves of female BALB/c mice (6–8 weeks old) that were challenged with $5 \times 10^5$ CFUs WT *C. albicans* after being vaccinated with the live or heat-killed *fas2Δ/Δ*, or mock-vaccinated control for durations of 15 days (E), 3 days (F), 28 days (G) or 2 months (H) (n = 8). Data are expressed as the mean ± SD in C and D. Statistical significance was determined using the log-rank test (B, E, F, G, and H), with *$P < 0.05$, **$P < 0.01$, ***$P < 0.001$ indicating significance levels, and ns denoting no significant difference.

## Discussion

In this study, we identified *FAS2* as a potential immunotherapeutic target against systemic *C. albicans* infection. Deletion of *FAS2* in *C. albicans* increases circulating innate immune responses and improve activated neutrophil fungicidal activity through unmasking immunogenic cell wall epitopes *via* the Mkc1-MAPK signaling pathway, leading to increased neutrophil fungicidal activity. Importantly, *FAS2* deletion does not impair major virulence factors or growth under serum-supplemented conditions, as *fas2Δ/Δ* retains normal yeast-to-hypha morphogenesis, growth, and intracellular fatty acid profiles. Furthermore, *fas2Δ/Δ* mutant elicits protective immune responses against subsequent systemic candidiasis in a murine model. Thus, our data indicate that *C. albicans FAS2* is a potential target for bolstering neutrophil-associated innate immunity. Since FASN-α is unique to fungi, targeting may provide a specific immunotherapeutic strategy for fungal infections without harming the host.

Furthermore, our findings align with the growing body of evidence that microbes can enhance host resistance through a mechanism known as "trained immunity" [43,44]. The *fas2Δ/Δ* mutant primes the host for a stronger, non-specific immune response to subsequent infections. This immune response is characterized by (i) single-cell sequencing revealing increased monocyte infiltration in the spleen without altering T and B cell counts; (ii) elevated circulating TNF-α, IL-1β, and IL-17; (iii) early protection against virulent *C. albicans* strain, evident within 7 dpi; and (iv) waning protection after two months. These features suggest a trained immunity phenotype, which enhances innate immune responses to re-encountered pathogens [45] and is distinct from adaptive immunity.

The role of neutrophils and macrophages, as professional phagocytes of the innate immune system, play a well-established role in combating fungal infections [2]. Recognition of β-glucan by the C-type lectin-like receptor Dectin-1 by these immune cells results in the release of inflammatory cytokines. Here, we demonstrate that *FAS2* deletion in *C. albicans* results in cell wall remodeling that exposes β (1,3)-glucan, a key trigger for cytokine release from innate immune cells. It is increasingly clear that TNF-α, a major pro-inflammatory cytokine produced by activated macrophages, plays an important role in protective anti-fungal responses [46]. We found that TNF-α production is also essential in response to *fas2Δ/Δ* infection. Yet, macrophage depletion does not negate the survival advantage, suggesting the involvement of additional mechanisms. We hypothesize that the exposed epitopes on *fas2Δ/Δ* mutants are more readily recognized by macrophages, prompting cytokine and chemokine production that recruits neutrophils for fungal clearance [47]. Since the efficacy of the immediate response during the first hours to days after fungal entry into the bloodstream determines whether fungal dissemination and growth can be contained [27,48], the rapid immune activation mediated by *fas2Δ/Δ* suggests the importance of circulating blood immune cells as a frontline defense against invasive infection. Further research is needed to elucidate the complex immune dynamics elicited by unmasked *C. albicans fas2Δ/Δ* cells.

The limited immune protection provided by heat-killed *fas2*Δ/Δ, despite the presence of β-glucan, suggests that additional factors may contribute to trained immunity. Emerging evidence points to fungal lipid metabolism as a potential modulator of immune training. In this study, *FAS2* deletion significantly altered fatty acid synthesis, potentially affecting immune recognition by modifying fungal cell wall structures or secondary metabolites [49]. Metabolic reprogramming, such as changes in fatty acid production, may influence immune responses, as observed with other fungal metabolites like lactate and formate, which are known to modulate immune training [50]. Thus, while β-glucan is crucial for trained immunity, our findings suggest *FAS2* deletion may reveal new lipid- or metabolite-based pathways influencing immune responses. Future research should aim to identifying these molecules and clarify their roles in immune modulation.

In light of these findings, the fungal FASN complex warrants reconsideration as a target for broad-spectrum antifungal drugs. However, the FASN-α subunit inhibitor, cerulenin, exhibited significantly reduced potency against *C. albicans* strains in the presence of human serum (S1 Table), indicating that its efficacy may be compromised *in vivo*. This limitation extends to other compounds targeting fatty acid synthesis, such as cerulenin (targeting FabB/FabH), triclosan (targeting FabI), or CG400549 (targeting FabI), which completely lose activity against Gram-positive pathogens like *S. agalactiae* in the presence of serum [51,52]. Additionally, antitumor agents that disrupt *de novo* fatty acid synthesis, including ND-646 (an allosteric inhibitor of acetyl-CoA carboxylase) [53] and multiple FASN inhibitors (such as C-75, orlistat, and GSK837149A), demonstrate transient effects and significant side effects [53,54]. These findings highlight the challenges of targeting the FASN complex for therapeutic purpose. Considering the influence of *in vivo* conditions on drug efficacy and the high costs of drug discovery, FASN-α might not be suitable for developing broad-spectrum antifungals.

Nevertheless, several key areas require further exploration to optimize *fas2*Δ/Δ as an immunotherapeutic target. First, it is important to continue identifying key immune cells in mediating protection, including dendritic cells in addition to macrophages and neutrophils. Indeed, during infection, hematopoietic tissue enhances the production and mobilization of immune cells such like neutrophils, which have short lifespans and must be produced in large numbers to fight infections [55]. Second, understanding how *fas2*Δ/Δ interacts with these cells and the associated signaling pathways, particularly those involved in trained immunity, could inform the development of targeted interventions to enhance long-term immunity. Third, given the transient nature of the immune protection observed in our study, strategies to prolong trained immunity warrant exploration. Approaches such as booster vaccinations or combining *fas2*Δ/Δ with immune modulators, could help sustain or amplify immune responses over time, ensuring more durable protection.

In summary, our data provide novel evidence that *C. albicans* FASN-α plays an essential role in regulating cell wall remodeling and the triggering of protective innate immunity, particularly during bloodstream infections. These findings suggest that targeting FASN-α could offer a promising strategy for antifungal immunotherapies. Future studies should explore how FASN-α-driven cell wall remodeling impacts the duration of trained immunity and explore potential synergies with existing antifungal treatments to improve the management of invasive fungal infections.

## Materials and methods

### Ethics statement

All animal experiments in this study were approved by the experimental animal ethics committee of Jinan University and were performed in accordance with the Guidelines for Care

and Use of Laboratory Animals of Jinan University (Approval No. IACUC-20231214-02) (Guangdong Province, China).

## Media and reagents

For routine propagation, the strains were grown in yeast peptone dextrose (YPD) broth or on YPD agar. YPD containing 200 μg/ml nourseothricin (Jena Biosciences) was used to select nourseothricin-resistant strains [56]. *Escherichia coli* DH5α was grown and maintained in Luria broth (LB). The following items were purchased from Biolegend (San Diego, CA): Anti-mouse CD45.2 antibody (104), anti-mouse CD11b (M1/70), and anti-mouse CD11c (N418). Anti-mouse Ly6G (IA8) was purchased from BioXcell. ELISA kits for mice cytokine were purchased from Multi Sciences (Hangzhou, China): IL-17A (EK217), TNF-α (EK282), IL-6 (EK206), and IL-1β (EK201B). Anti-mouse β (1,3)-glucan antibody and goat anti-mouse IgG (H+L) Alexa Fluor 488 secondary antibody were purchased from Abcam (ab233743, ab150113). Anti-rabbit phospho-PKC and anti-rabbit phospho-p44/42 MAPK (Thr$^{202}$/Tyr$^{204}$) antibodies were purchased from Cell Signaling Technology (6967S, 4370T). The mouse anti-tubulin primary antibody was purchased from Beijing Ray Antibody Biotech. Wheat Germ Agglutinin (WGA, 25530) was obtained from AAT Bioquest. Calcofluor white (CFW, Sigma, 18909) and Concanavalin A (ConA, C7642) were obtained from Sigma-Aldrich. The validation of the antibodies used is provided on the manufacturers' websites.

## Generation of the gene deletion mutant and complementation strain

The strains employed in this study were included in S2 Table. *fas2Δ/Δ* and *fas2Δ/FAS2* strains were generated in strain SC5314 using the *SAT1*-Flipper method [56,57]. A list of all plasmids, and primers for strain construction used in this study can be found in S3 and S4 Tables, respectively. Briefly, we amplified approximately 600 bp of the 5' promoter and 3' terminator regions adjacent to the *FAS2* open reading frame (ORF) with specific primer pairs, digested them with appropriate restriction enzymes, and cloned these into the *SAT1*-flipper plasmid pSFS2. The modified plasmid pSFS2-*FAS2* was linearized and transformed into the SC5314 strain, with selection on YPD medium supplemented with 200 μg/ml nourseothricin (YPD-Nou). Transformants were grown in YPM medium with maltose to induce caFLP expression, resulting in the *fas2Δ/FAS2* heterozygous mutant after confirming the deletion with PCR. The process was repeated to delete the second *FAS2* allele in the *fas2Δ/FAS2* heterozygous strain. The plasmid pSFS2-*FAS2* was again linearized and transformed, with selection on YPD-Nou, and PCR was used to verify correct integration and *FAS2* ORF deletion. To create the *fas2Δ/Δ* homozygous mutant, Mal2pFLP recombinase expression was induced by growth on yeast extract-peptone-maltose medium, which restored nourseothricin sensitivity. For the *FAS2*-restored strain, we complemented the mutant with a plasmid containing the complete *FAS2* ORF and its regulatory regions, cloned into pSFS2-down, and transformed into the *fas2Δ/Δ* mutant. Cells were then cultivated in YPD with or without fatty acids. If not otherwise mentioned, the strains were grown in either YPD (1% yeast extract, 2% bactopeptone, 2% glucose), or YPDT40 (YPD plus 1% Tween 40, 0.01% myristic and stearic acids) (17). Schematic diagram of the *fas2Δ/Δ* and *fas2Δ/FAS2* complemented strain, and confirmation of the *fas2Δ/Δ* mutant, *fas2Δ/FAS2* complemented strain was shown in S7 Fig.

## Growth conditions

For the *in vitro* growth assay, *C. albicans* strains were propagated in YPDT40 broth and grown with shaking at 30 °C overnight. Then, Cells were inoculated in 100 ml of YPD, yeast extract peptone succinate (YPS), or YPD plus 50% filtered human serum broth adjusted to an initial

absorbance of 0.02 at 600 nm using a Varioskan LUX Multimode Microplate Reader (Thermo Scientific) and grown with shaking at 30 °C. At the indicated time points, samples were taken to measure the absorbance at 600 nm, and appropriate dilutions of each sample were plated on duplicate on YPD plates to determine the number of CFU. For the *ex vivo* growth assay, the kidneys, livers, and spleens were removed aseptically from the mice. The organs were cut into small pieces by using sterile scissors and forceps. An even suspension of the tissue was prepared with a sterile tissue grinde using sterile saline solution (1 g of tissue: 2 ml of saline solution). The suspension was poured into a large, screw-capped test tube, using small amounts of saline solution to rinse all particles from the grinder. The suspension was then placed onto a sterile 6-well plate (Corning), and the homogenate was inoculated with *C. albicans*. At the indicated time points, samples were plated on YPD agar for 48 h to determine the CFU/ml.

### *In vivo* virulence assay

Female BALB/c mice (6–8 weeks old) were housed under controlled conditions with a temperature of 20–26 °C, humidity of 40–70%, and a 12/12-hour dark/light cycle. The indicated *C. albicans* strains were harvested from overnight cultures, washed twice, and resuspended in PBS. Mice were intravenously infected with $5 \times 10^5$ CFU of WT, $5 \times 10^5$ CFU of *fas2Δ/Δ*, $5 \times 10^5$ CFU of *fas2Δ/FAS2*, or $5 \times 10^6$ CFU of *fas2Δ/Δ* or inoculation with 100 µl of saline as a control. Mouse survival was monitored and recorded daily. To reveal in vivo fungal morphology and tissue damage, kidneys were removed aseptically from mice 3 days post-infection and fixed in 10% neutral buffered formalin, embedded in paraffin, and sections were stained with periodic acid-stained (PAS).

### Systemic priming

Female BALB/c mice (6–8 weeks old) were immunized via tail vein injection of $5 \times 10^5$ CFUs of *C. albicans* (WT or *fas2Δ/Δ*) strains. A secondary systemic challenge with a lethal dose ($5 \times 10^5$ CFUs) of SC5314 was performed at 3, 7, 15, 28, or 60 days after the primary immunization as specified in different experimental protocols. Mouse survival was monitored and recorded daily in all experiments.

### Single-cell RNA sequence and cell type annotation

Spleen tissues were collected from mice infected with *C. albicans* strains for 24 h. Single cells were digested from tissues as described previously [58]. Singleron Biotechnologies provides technical services. For primary analysis of raw read data, raw reads were processed to generate gene expression profiles using CeleScope v1.5.2 with default parameters. Briefly, Barcodes and UMIs were extracted from R1 reads and corrected. Adapter sequences and poly-A tails were trimmed from R2 reads and the trimmed R2 reads were aligned against the GRCh38 (hg38) {GRCm38 (mm10)} transcriptome using STAR (v2.6.1b). Uniquely mapped reads were then assigned to genes with FeatureCounts (v2.0.1). Successfully Assigned Reads with the same cell barcode, UMI, and gene were grouped to generate the gene expression matrix for further analysis. Scanpy v1.8.2 was used for quality control, dimensionality reduction, and clustering under Python 3.7. Cells were separated into 9 clusters using the Louvain algorithm, setting the resolution parameter at 1.2. Cell clusters were visualized using Uniform Manifold Approximation and Projection (UMAP). The batch effect between samples was removed by Harmony v1.0 using the top 20 principal components from PCA.

Cell-ID is a multivariate approach that extracts gene signatures for each cell and performs cell identity recognition using hypergeometric tests (HGT). Dimensionality reduction was performed on a normalized gene expression matrix through multiple correspondence analysis,

where both cells and genes were projected in the same low-dimensional space. Then a gene ranking was calculated for each cell to obtain the most featured gene sets of that cell. HGT was performed on these gene sets against spleen reference from the SynEcoSys database. The identity of each cell was determined as the cell type has the minimal HGT p-value. For cluster annotation, the frequency of each cell type was calculated in each cluster, and the cell type with the highest frequency was chosen as the cluster's identity.

The cell type identification of each cluster was determined according to the expression of canonical markers from the reference database SynEcoSys (Singleron Biotechnology). To obtain a high-resolution map of MPs and neutrophils from the specific cluster were extracted and reclustered for more detailed analysis following the same procedures described above and by setting the clustering resolution as 0.8.

## Detection of serum and kidney cytokines by ELISA

For detection of TNF-α, IL-6, and IL-1β in macrophages and neutrophils cultures supernatants, $1 \times 10^5$ BMDMs or BM-neutrophils were infected with *C. albicans* cells at a MOI of 1:1 for the time indicated, and cytokine production in the supernatant was measured by ELISA. For detection of serum TNF-α, IL-6, IL-17A, and IL-1β, mice infected with *C. albicans* (WT, *fas2Δ/Δ*, and *fas2Δ/FAS2*), sera were collected at indicated time points and subjected to ELISA analysis. The kidneys harvested at the indicated time after infection were homogenized, and the supernatant was recovered following centrifugation at 15,000 *g* for 20 min at 4 °C. ELISA kits determined the cytokines, including TNF-α, IL-6, IL-17A, and IL-1β according to the manufacturer's instructions. The ELISA results were expressed as 'pg per g of kidney'.

## Generation of BMDMs and isolation of mouse BM neutrophils

BM cells were harvested from the femurs and tibias of mice. Cells were cultured in Dulbecco's modified Eagle's medium (DMEM, Gibco) containing 10% heat-inactivated fetal bovine serum (FBS, Invitrogen) with 20 ng/mL macrophage colony-stimulating factor (M-CSF). After one week of culture, non-adherent cells were removed, and adherent cells were 80–90% F4/80+CD11b+, as determined by flow cytometric analysis. To isolate BM neutrophils, total BM cells were recovered from the femurs and tibias by flushing with RPMI-1640 medium (Gibco) with an 18-gauge needle. Erythrocytes were lysed with red blood cell (RBC) lysis buffer (Sigma-Aldrich) and BM neutrophils were isolated by percoll from Solarbio (P8370) according to the manufacturer's protocol.

## Flow cytometry

Antibody labeling of immune cells was carried out in FACS staining buffer (PBS supplemented with 2% PBS and 2 mM EDTA) on ice for 30 min. Immune cell recruitment was assessed by collagenase 4 (Worthington) and DNase (Roche) digestion of organs, RBC lysis, antibody staining adding counting beads, and subsequent flow cytometric analysis. Samples were acquired with a FACSCanto flow cytometer (BD Biosciences), and data were analyzed using FlowJo software.

## *C. albicans*-macrophage co-culture assays

RAW264.7 cells were seeded onto a 35 mm glass plate and stained with 500 nM MitoTracker Deep Red FM (Molecular Probes) for 30 minutes to visualize the mitochondria. *C. albicans* cells were labeled with a FITC fluorescent dye from Sigma Aldrich at a concentration of 1.25 mM for 15 minutes at room temperature, followed by three washes to remove any excess dye. The FITC-labeled *C. albicans* and macrophages were then co-cultured at a MOI of 2.

After co-culture, the cells were washed twice with PBS, and the samples were examined using a CLSM (Carl Zeiss LSM 880) with specific excitation and emission wavelengths for macrophages (Ex 644/Em 655) and *C. albicans* (Ex 488/Em 525). Imaging was performed on a Leica TCS-SP2 confocal microscope (1:100).

### Filamentation assays

Filamentation of *C. albicans* strains was assayed by incubation in 10% FBS, Spider, or Lee's liquid media at 37 °C in 12-well flat-bottomed presterilized microtiter plate for 2 h. The plates were visualized under an inverted microscope and photographed.

### LC-MS untargeted metabolomics

This study utilized Liquid Chromatography-Mass Spectrometry (LC-MS) for untargeted metabolomic analysis to comprehensively identify and quantify metabolites in *C. albicans* cells. Shanghai Luming Biotechnology Co., Ltd. provides technical services. After appropriate preprocessing and chromatographic separation of the samples, they were ionized and analyzed by the mass spectrometer. A liquid chromatography-mass spectrometry system composed of an ACQUITY UPLC I-Class plus ultra-high performance liquid chromatography-tandem QE high-resolution mass spectrometer is utilized for this analysis. The resulting mass spectrometry data were processed and compared using Progenesis QI v2.3 software to identify and quantify differential metabolites in the samples.

### Macrophage cytotoxicity assay

To evaluate the damage induced by *C. albicans* on macrophages, the release of lactate dehydrogenase (LDH) was measured using a non-radioactive cytotoxicity assay kit from Promega. RAW264.7 cells and *C. albicans* were co-cultured in 96 well plates at respective densities of $1 \times 10^5$ cells/well and $2 \times 10^5$ CFU/well. The co-cultures were incubated for durations of 6, 12, and 24 hours. At each time point, the plate was centrifuged at $250 \times \boldsymbol{g}$ for 5 minutes, and 50 μl of the supernatant was transferred to a new 96-well plate. This was then mixed with 50 μl of the substrate mixture and incubated for 30 minutes. The reaction was halted by the addition of 50 μl of termination solution, and the absorbance at 490 nm was measured and recorded, providing a quantitative assessment of LDH release.

### Immune cell depletion in mice

Non-specific immune cell depletion was achieved by recurring injections with cyclophosphamide (Sigma-Aldrich). Female BALB/c mice (6–8 weeks old) were weighed prior to treatment to determine the average weight of all mice. Mice then were treated with 150 mg/kg of cyclophosphamide *via* intraperitoneal (I.P.) injections starting 4 days prior to infection. Immune depletion was then maintained with recurring I.P. injections of 150 mg/kg of cyclophosphamide every 3 days until the experiment was terminated.

*In vivo* neutrophil depletion was accomplished using an anti-mouse Ly6G (IA8) monoclonal antibody (BioXcell). Mice were intraperitoneal injected with 300 μg of the IA8 antibody starting 1 day prior to infection, and neutrophil depletion was maintained via recurring I.P. injections of 300 μg of IA8 every other day until the end of the experiment.

For clodronate- or PBS-liposome treatment experiments, mice were given I.P. injections with 100 μl per 10 g mouse weight of liposomes both 24 hours before and after intravenous infection with *C. albicans*. The clodronate and PBS control liposomes were acquired from http://clodronateliposomes.org and handled according to the manufacturer's instructions [30].

## RT-qPCR

cDNA of *C. albicans* strains was synthesized using the PrimeScript RT Master Mix (ABKbio). All quantitative real-time PCR (RT-qPCR) was conducted in triplicate using SYBR Green qPCR SuperMix (ABKbio), and the amplified product was monitored with a MiniOpticom Real-time System (Bio-Rad). 18 S rRNA was the housekeeping gene for normalization.

Mice infected with *C. albicans* strains for indicated times were sacrificed. The cells from the kidney homogens were harvested and counted. Total RNA was isolated using 1 ml of Trizol (Invitrogen) according to the manufacturer's protocol. RNA (500 ng) was reverse transcribed with PrimerScript RT Master Mix. RT-qPCR was performed using SYBR Green qPCR SuperMix. GAPDH served as the housekeeping gene. The primers for the genes used in this study are shown in S4 Table. Data analysis was performed using the $2^{-\Delta\Delta CT}$ method [30].

## Rho1 activity assay

Utilizing the Rho Activation Assay Biochemistry Kit provided by Cytoskeleton (Catalog No. BK036), we conducted an assay to gauge the activity of the Rho1 protein. In alignment with the protocol furnished by the kit's manufacturer, *Candida albicans* from each experimental group, during the logarithmic growth phase, were cultivated in 50 milliliters of YPD medium until the optical density at 600 nanometers ($OD_{600}$) reached a value of 0.1. The cells were harvested, rinsed with chilled PBS to remove debris, and then subjected to mechanical disruption using liquid nitrogen. The cell lysate was prepared to extract the total protein content. The protein concentration was ascertained using the BSA (Bovine Serum Albumin) standard method, ensuring accurate quantification. A precise amount of 1 mg of the extracted protein was then allowed to interact with the Rhotekin RhoA-binding domain beads at 4 °C for an hour to facilitate binding. Following the incubation period, the beads are washed once with the provided washing buffer, and 2 × Laemmli sample buffer is added. The samples were then subjected to western blot analysis, employing a mouse monoclonal anti-RhoA antibody to specifically detect the Rho1 protein.

## Western blotting analysis

Western blotting was conducted following established methods. The phosphorylation of PKC and Mkc1 MAPK was detected using anti-rabbit phospho-PKC and anti-rabbit phospho-p44/42 antibodies from Cell Signaling Technology, at a dilution of 1:1000. A secondary goat anti-rabbit IgG-HRP conjugate from Xiamen Tagene Biotechnology (TA005) was utilized for phospho-p44/42, and phospho-PKC antibodies. As a loading control, beta-tubulin was detected using a mouse anti-tubulin primary antibody from Beijing Ray Antibody Biotech, at a 1:5000 dilution, followed by a secondary goat anti-mouse IgG-HRP conjugate (Xiamen Tagene Biotechnology, TA006). After imaging, the ratios of phosphorylated PKC and Mkc1-MAPK to beta-tubulin were quantified by densitometry using the Odyssey Imager from Li-Cor Biosciences, with image analysis conducted in ImageJ software. The quantification was based on three biological replicates per sample.

## Measurement of nitric oxide concentration

Cell culture supernatants were collected, and the concentration of nitric oxide (NO) was measured using an NO assay kit (S0021) from Beyotime Biotechnology based on the Griess reaction. The NO concentration was ascertained by reference to a standard curve established with sodium nitrite (NaNO2), and all samples were prepared according to the manufacturer's protocol.

## NET formation assay

The confocal microscope was used to visualize NET formation, neutrophils ($5 \times 10^5$ cells) were stimulated with live *C. albicans* (MOI of 1:10) at 37 °C, and the cells were stained overnight at 4°C with anti-myeloperoxidase (Proteintech, 22225-1-AP, diluted 1:100). Two more washes were made, followed by addition of secondary antibodies goat anti-mouse IgG (H+L) Alexa Fluor 488 (ab150113, diluted 1:50) and incubated for 1 h at 37 °C. Next, cells were stained by DAPI for 15min. Epifluorescence images were captured using an Olympus FV3000 Laser-Scanning Confocal Spectral Inverted Microscope. The pictures were taken and analyzed through Olympus FV31S-SW V2.6 software. Quantification of microscopy images was achieved with the use of ImageJ.

## ROS assay, phagocytosis of *C. albicans* and fungal killing assay

For the ROS production assay, $5 \times 10^5$ BMDMs or neutrophils were washed with PBS twice, cells were incubated at 37 °C for 2 h and then infected with *C. albicans* at a MOI of 2:1. The relative amounts of ROS generated by BMDMs or neutrophils were detected by using a ROS Assay Kit from Beyotime Biotechnology (S0033S).

For phagocytosis of *C. albicans*, *C. albicans* were labeled with Alexa Fluor 488 (Invitrogen) in 100 mM HEPES buffer (pH 7.5) (diluted to 1:500) and then co-cultured with BMDMs or neutrophils for the indicated times at 37 °C. Adherent fungal cells were quenched with trypan blue, and phagocytosis was determined by the Varioskan LUX multimode microplate reader.

For *in vitro* fungal killing assay, BMDMs or neutrophils ($1 \times 10^5$/well) were incubated with *C. albicans* at a MOI of 1:500 for 24 h. After co-culture, a 100 μl suspension was spread on YPD plates. After incubation at 37 °C for 48 h, killing was determined by counting the *Candida* colonies, with or without the indicated cells.

## Transmission electron microscope assay

*C. albicans* strains were cultivated overnight at 30°C in YPD medium, then transferred to YNB medium supplemented with 2% glucose and incubated for 12 hours. The cells were then washed with PBS, pelleted, and fixed overnight in a fixative solution containing 4% paraformaldehyde and 2.5% glutaraldehyde. The fixed cells were stored at 4°C until further processing for electron microscopy. For postfixation, the cells were treated with 1% osmium tetroxide. Subsequently, they underwent a series of dehydration steps in ethanol solutions of increasing concentrations (15-minute washes in 10%, 25%, 40%, 55%, 70%, 85%, and 100% ethanol). The dehydrated cells were then embedded in epoxy resin following standard procedures. Ultrathin sections, 70 nanometers in thickness, were prepared from the embedded samples and examined using a transmission electron microscope (TEM) model JEOL JEM-1230.

## Staining of cell wall components

For staining of fungal cells, 5 ml of *C. albicans* strains in YPD were started the morning before staining. After overnight growth, cells were diluted to an $OD_{600}$ of 0.5. Cells were then cultured overnight at 4°C with a mouse anti-β (1,3)-glucan antibody (ab233743, diluted 1:100). Subsequently, a secondary antibody, goat anti-mouse IgG (H+L) Alexa Fluor 488 (ab150113, diluted 1:50), was applied to enhance the staining process. Total chitin was assessed by staining cells with 500 μl of a 10 μg/ml calcofluor white solution in PBS for 5min. To assess exposed chitin, cells were stained with 500 μl of a 10 μg/ml fluorescein-labeled wheat germ agglutinin (WGA) (AAT Bioquest) solution in PBS for 30min. Mannan levels were assessed by staining cells with 500 μl of a 50 μg/ml solution of Alexa Fluor 488 conjugated concanavalin A (Sigma) in PBS

and were incubated with shaking at 4 °C for 30 min. At least 3 biological replicates consisting of 10,000 recorded events each were used to assess all strains by flow cytometry.

After immunofluorescent staining of yeast cells for exposed β (1,3)-glucan, mannan, and chitin, the samples were observed under an Olympus FV3000 Laser-Scanning Confocal Spectral Inverted Microscope, and the images were taken through Olympus FV31S-SW V2.6 software.

### Preparation of monoclonal antibodies

A rabbit anti-FASN-α subunit antibody was produced by Guangzhou Lide Biotech. Briefly, the peptides are diluted in a coating solution and applied to an ELISA plate for antigen coating, enabling the detection of the specific antibody via ELISA. The immunogen is prepared by coupling peptides with keyhole limpet hemocyanin (KLH) using the bifunctional anomeric linker SMCC (succinimide 4-[N-methyl] cyclohexane-1-carboxylate), forming a (peptide-SMCC-KLH) complex. Two healthy New Zealand white rabbits receive a primary and booster immunization schedule tailored to the immunogen's properties. Seven days after the immunogen injection, blood is drawn, centrifuged at 3500 rpm for 15 minutes, and the serum is collected. This serum is stored in sterilized cryovials with 0.02% sodium azide at -20 °C after two rounds of centrifugation. The antibody-rich serum is purified using an affinity chromatography column, and the purified antibodies are collected, dialyzed, and concentrated for subsequent use.

### Statistical analysis

Before performing statistical analyses, GraphPad Prism 9.0 software was used to test the normal distribution of each dataset. Based on the distribution (normal or not normally distributed) the appropriate statistical test was chosen and performed as described in each figure caption. Statistical tests and *P* values are described in the manuscript and figure legends. *P* < 0.05 was considered significant.

### Supporting information

**S1 Fig. *FAS2* is required for pathogenicity.** (A, B) Survival curves (A) and weight loss (B) of mice (n = 8) after intravenous injection with *C. albicans* WT, *fas2Δ/Δ*, *fas2Δ/FAS2* at $5 \times 10^5$ CFUs per mice, and *fas2Δ/Δ* at $5 \times 10^6$ CFUs per mice. (C, D) Representative images of PAS stained kidney sections of mice (C) (n = 3) 72 h after systemic infection with WT, *fas2Δ/Δ* and *fas2Δ/FAS2*, and combined inflammatory score based on renal immune cell infiltration and tissue destruction (n = 3) after intravenous infection with *C. albicans* (D). (E-H) *C. albicans* fungal burdens (CFU per g of tissue) in organs at 1, 24, 48 and 72 h postinfection with WT, *fas2Δ/Δ* and *fas2Δ/FAS2* ($5 \times 10^5$ CFUs per mice) (n = 3). Data in (A, B, D, E, F, and H) are presented as mean ± SD. Statistical significance was determined using the log-rank test for survival curves (A), two-tailed unpaired Student's *t*-test for inflammatory score (D), and two-way ANOVA for fungal burdens (E, F, G, and H). Significance is indicated as *P < 0.05, **P < 0.01, ***P < 0.001, with ns denoting no significant difference.
(TIF)

**S2 Fig. The deletion of *FAS2* leads to remarkable growth defects in *C. albicans*, but normal morphogenesis under *in vitro* hypha-inducing conditions.** (A, B) The growth of WT, *fas2Δ/Δ* and *fas2Δ/FAS2* in YPD (A, B), or YPS (C) media over time determined by cell density ($OD_{600}$) (A, C) and viable cell counts (CFU/ml) (B). (D-F) Hyphae formation of WT, *fas2Δ/Δ* or *fas2Δ/FAS2* strains was induced in Spider and 10% FCS liquid media for 2 h at

37 °C (D). In D magnification × 400. The percentage of hyphal cells (E) was calculated at least 100 cells in each group (n = 3). (F) The mRNA expression levels of the hyphae-associated genes of WT, *fas2Δ/Δ* or *fas2Δ/FAS2* cultured in 10% FCS medium for 6 h as assessed by RT-qPCR. Data are expressed as the mean ± SD of three independent experiments. Statistical significance is indicated by *$P$ < 0.05, **$P$ < 0.01, ***$P$ < 0.001, with 'ns' for not significant. Two-way ANOVA was used for statistical analysis in A-C; In D, one representative experiment out of three independent experiments is shown; the two-tailed unpaired Student's t-test for E, and F.
(TIF)

**S3 Fig. Evaluation of the immune responses after infection with *fas2Δ/Δ*.** (A) Peripheral blood of mice was analyzed at 24, 48, 72 h, and 6 d after intravenous infection with *C. albicans* WT, *fas2Δ/Δ*, and *fas2Δ/FAS2* ($5 \times 10^5$ CFUs per mice) using FACS, with cells stained with antibodies to CD11b, CD11c, Ly6c and Ly6G. One representative FACS plot per group (n = 3) was shown. (B) Kidney cells of mice were analyzed for immune cell populations at 24, 48, 72 h, and 6 d after intravenous infection with *C. albicans* WT, *fas2Δ/Δ*, and *fas2Δ/FAS2* ($5 \times 10^5$ CFUs per mice) using FACS stained with antibodies to CD11b, CD11c, Ly6c and Ly6G (n = 3), and one representative experiment of three independent experiments is shown. (C, D) Representative images of TUNEL staining of mice infected with WT, *fas2Δ/Δ* and *fas2Δ/FAS2* (C), and quantification of TUNEL-positive cells (D). (E) Kidneys were isolated for RT-qPCR analysis of indicated acute kidney injury genes or adhesion molecules expression after intravenous infection with *C. albicans* WT, *fas2Δ/Δ*, and *fas2Δ/FAS2* ($5 \times 10^5$ CFUs per mice) (n = 3). Significance is indicated as *$P$ < 0.05, **$P$ < 0.01, ***$P$ < 0.001, with ns denoting no significant difference.
(TIF)

**S4 Fig. Immune cell composition and sub-cluster distribution in spleens of WT, *fas2Δ/Δ*, and mock-infected mice.** (A) UMAP plots depicting the clustering of splenic immune cells from WT, *fas2Δ/Δ*, and mock-infected mice. Each dot represents a single cell, colored according to its assigned cell type, with major populations including splenocytes, neutrophils, and monocyte phagocytes. Cells are partitioned into nine, seven, and eight distinct populations in WT, *fas2Δ/Δ*, and mock groups, respectively. (B) Stacked bar plot showing the relative abundance of monocyte phagocyte sub-clusters across WT, *fas2Δ/Δ*, and mock-infected samples (n = 3), with populations such as dendritic cells (cDC1, cDC2), macrophages, and classical/non-classical monocytes. (C) Stacked bar plot illustrating the proportion of neutrophil sub-clusters in the three groups (n = 3), categorized by marker genes (*Ifitm6*, *Cers6*, *Ccl6*, *Ccrl2*, *Retnlg*, *Ltf*, and *Elane*). (D-E) Heatmaps displaying the expression levels of signature genes for splenic immune cell clusters (D), monocyte phagocyte sub-clusters (E), and neutrophil sub-clusters (F) across WT, *fas2Δ/Δ*, and mock samples. The color intensity represents the expression level, highlighting differences in gene expression among specific clusters and sub-clusters between the groups.
(TIF)

**S5 Fig. Macrophages are required for the *fas2Δ/Δ* mutant induced protective immunity.** (A, B) Phagocytosis and killing capacity of BMDMs after infecting WT, *fas2Δ/Δ* and *fas2Δ/FAS2* at a MOI=1 (n = 3). (C-E) Bone marrow-derived macrophages (BMDMs) from BALB/c mice were cultured with WT, *fas2Δ/Δ* and *fas2Δ/FAS2* at a MOI=1 for 2 h followed by measurement of the proinflammatory cytokines levels of TNF-α (C) and IL-1β (D) in cell supernatants by ELISA (n=3), and gene expression assessment by RT-qPCR (n = 3) (E). (F) BMDMs were co-cultured with WT, *fas2Δ/Δ* and *fas2Δ/FAS2*, followed by RT-qPCR analysis of MPO, S100a8, and iNOS transcript levels (n = 3). (G) BMDMs were stimulated WT, *fas2Δ/Δ*, and *fas2Δ/FAS2* for 2 h, followed by measurement of ROS production via a fluorescent ROS probe

and the MFI analysis (n = 3). (H) RAW 264.7 macrophages preloaded with MitoTracker Deep Red FM (red) were co-cultured with FITC-stained WT, *fas2Δ/Δ* and *fas2Δ/FAS2* cells (green). Magnification ×120. (I) The images of pretreated RAW264.7 macrophages and WT, *fas2Δ/Δ* or *fas2Δ/FAS2* strains after 12 h co-culture at a MOI of 1:1 ratio were captured by the Leica DMi8 microscope. These images represent one of three separate experiments. (J) Survival rates of WT, *fas2Δ/Δ* and *fas2Δ/FAS2* cells by the co-cultured with RAW264.7 macrophages. Macrophages within 24 h were determined by the endpoint dilution assay at each time point. (K) Macrophage cytotoxicity caused by WT, *fas2Δ/Δ* and *fas2Δ/FAS2* cells were determined by the release of LDH. (L, M) Following the co-culture of RAW264.7 cells with WT, f*as2Δ/Δ*, and *fas2Δ/FAS2* strains at a MOI of 1:1, the levels of intracellular nitric oxide (NO) and the expression of iNOS were measured. Significance is indicated as $*P < 0.05$, $**P < 0.01$, $***P < 0.001$, with ns denoting no significant difference.
(TIF)

**S6 Fig.  Impact of the *fas2Δ/Δ* mutant on fungal cell wall structure.** (A) Growth of WT, *fas2Δ/Δ* and *fas2Δ/FAS2* exposed to cell wall-perturbing agents. Serial 10-fold dilutions of the cells were spotted onto YPD plates in the presence of the following agents: congo red (CR), calcofluor (CFW) and caspofungin (CAS). The plates were incubated for 48 h at 30 °C. (B) WT, *fas2Δ/Δ* and *fas2Δ/FAS2* strains were grown to mid-log phase and incubated with recombinant β(1,3)-glucanase. The decrease in $OD_{600}$ represents cell lysis as the β(1,3)-glucanase digests the cell wall and is expressed as a percentage of the starting $OD_{600}$. (C, D) Quantification of Aniline Blue by FACS analysis counting 10,000 events per repeat (C). The MFI from three independent experiments is displayed with standard deviation (D). Statistical significance is denoted as $**P < 0.01$ and $***P < 0.001$, with 'ns' indicating no significance.
(TIF)

**S7 Fig.  Validation of *FAS2* gene deletion and complementation.** (A-D) The *FAS2* gene was PCR-amplified from the genomic DNA of the WT, *fas2Δ/Δ* and *fas2Δ/FAS2* with primers FAS2-F+FAS2-R. WT was an amplified stripe at 1156 bp; *fas2Δ/Δ* was without an amplified stripe; *fas2Δ/FAS2* amplified stripe at 1156 bp. (E) RT–qPCR analysis of the mRNA level in *FAS2* from the three strains was performed. (F) The FASN-α subunit level in WT, *fas2Δ/Δ* and *fas2Δ/FAS2* was assessed by western blot using an anti-FAS1 antibody (provided by Guangzhou Lide Biotech). (G) Quantification of FASN-α subunit to the β-Tubulin fraction in WT, *fas2Δ/Δ* and *fas2Δ/FAS2* strains, represented as Fold-change (n = 3 biological replicates across 3 separate blots). In E, and G, data are expressed as the mean ± SD of three independent experiments. Statistical significance is indicated by $**P < 0.01$ and $***P < 0.001$, with 'ns' denoting no significance, as determined by the two-tailed unpaired Student's *t*-test.
(TIF)

**S1 Table.  Cerulenin and S-104 inhibition in the presence of human serum.**
(XLSX)

**S2 Table.  Strains used in this study.**
(XLSX)

**S3 Table.  Plasmids used in this study.**
(XLSX)

**S4 Table.  Primers used in this study.**
(XLSX)

**S1 Data. A list of relative proportions of major cell types in single-cell sequencing samples from WT, *fas2Δ/Δ*, and mock-infected mice.**
(XLSX)

**S2 Data. A list of relative proportions of mononuclear phagocytes in single-cell sequencing samples from WT, *fas2Δ/Δ*, and mock-infected mice.**
(XLSX)

**S3 Data. A list of relative proportions of neutrophils in single-cell sequencing samples from WT, *fas2Δ/Δ*, and mock-infected mice.**
(XLSX)

**S4 Data. A list of significantly differential lipidomic metabolites among WT, *fas2Δ/Δ*, and *fas2Δ/FAS2*.**
(XLSX)

**S5 Data. Enriched metabolic pathway for the differentially abundant lipidomic metabolites in WT versus *fas2Δ/Δ*.**
(XLSX)

**S6 Data. A list of significantly differential metabolites among WT, *fas2Δ/Δ*, and *fas2Δ/FAS2* was identified through untargeted metabolomics analysis.**
(XLSX)

**S7 Data. A list of the top 30 differential metabolites in WT and *fas2Δ/Δ* screed by untargeted metabolomics.**
(XLSX)

**S8 Data. Excel spreadsheet containing the underlying numberical data in separate sheets for Figure panels.**
(XLSX)

## Acknowledgements

We would like to thank Mr. Xinqiang Lai from the Analytical and Testing Center of Jinan University for his instructions on flow cytometry analysis.

## Author contributions

**Conceptualization:** Yajing Zhao, Hong Zhang.

**Data curation:** Zhishan Zhou, Guiyue Cai, Dandan Zhang, Xiaoting Yu, Dongli Zhang, Jiyao Luo.

**Formal analysis:** Yajing Zhao, Zhishan Zhou, Guiyue Cai, Dandan Zhang.

**Funding acquisition:** Yajing Zhao, Aili Gao, Hong Zhang.

**Investigation:** Zhishan Zhou, Guiyue Cai, Dandan Zhang, Xiaoting Yu, Dongli Zhang, Jiyao Luo, Yunfeng Hu.

**Methodology:** Yajing Zhao, Zhishan Zhou, Guiyue Cai, Dongmei Li, Shuixiu Li, Zhanpeng Zhang.

**Project administration:** Yajing Zhao.

**Resources:** Yajing Zhao, Hong Zhang.

**Supervision:** Yajing Zhao, Shuixiu Li, Zhanpeng Zhang.

**Validation:** Zhishan Zhou, Guiyue Cai, Dandan Zhang, Aili Gao.

**Visualization:** Yajing Zhao, Zhishan Zhou, Guiyue Cai.

**Writing – original draft:** Yajing Zhao, Guiyue Cai, Dandan Zhang, Dongmei Li.

**Writing – review & editing:** Yajing Zhao, Guiyue Cai, Aili Gao, Hong Zhang.

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
