## [Decision Letter · Decision Letter 0]

11 Dec 2024

PPATHOGENS-D-24-02439

Systemic infection by Candida albicans  requires FASN-α subunit induced cell wall remodeling to perturb immune response

PLOS Pathogens

Dear Dr. Zhao,

Thank you for submitting your manuscript to PLOS Pathogens. After careful consideration, we feel that it has merit but does not fully meet PLOS Pathogens's publication criteria as it currently stands. Therefore, we invite you to submit a revised version of the manuscript that addresses the points raised during the review process.

Please submit your revised manuscript within 30 days Feb 09 2025 11:59PM. If you will need more time than this to complete your revisions, please reply to this message or contact the journal office at plospathogens@plos.org. Please include the following items when submitting your revised manuscript:

We look forward to receiving your revised manuscript.

Kind regards,

David R. Andes

Academic Editor

PLOS Pathogens

Michal Olszewski

Section Editor

PLOS Pathogens

Sumita Bhaduri-McIntosh

Editor-in-Chief

PLOS Pathogens

orcid.org/0000-0003-2946-9497

Michael Malim

Editor-in-Chief

PLOS Pathogens

orcid.org/0000-0002-7699-2064

**Additional Editor Comments :**

The reviewers appreciate the response to prior reviews and manuscript edits. A few issues remain that we are hopeful you will be able to address.

**Journal Requirements:**

At this stage, the following Authors/Authors require contributions: Yajing Zhao, Zhishan Zhou, Guiyue Cai, Dandan Zhang, Xiaoting Yu, Dongmei Li, Shuixiu Li, Zhanpeng Zhang, Dongli Zhang, Jiyao Luo, Yunfeng Hu, Aili Gao, and Hong Zhang. Please ensure that the full contributions of each author are acknowledged in the "Add/Edit/Remove Authors" section of our submission form.

https://journals.plos.org/plospathogens/s/submission-guidelines#loc-parts-of-a-submission

3) We noticed that you used the phrase 'unpublished ' in the manuscript. We do not allow these references, as the PLOS data access policy requires that all data be either published with the manuscript or made available in a publicly accessible database. Please amend the supplementary material to include the referenced data or remove the references.

4) We do not publish any copyright or trademark symbols that usually accompany proprietary names, eg ©,  ®, or TM  (e.g. next to drug or reagent names). Therefore please remove all instances of trademark/copyright symbols throughout the text, including:

- TM on page: 40.

5) Please upload all main figures as separate Figure files in .tif or .eps format. For more information about how to convert and format your figure files please see our guidelines: 

6) We have noticed that you have uploaded Supporting Information files, but you have not included a list of legends. Please add a full list of legends for your Supporting Information files after the references list.

7) Some material included in your submission may be copyrighted. According to PLOS copyright policy, authors who use figures or other material (e.g., graphics, clipart, maps) from another author or copyright holder must demonstrate or obtain permission to publish this material under the Creative Commons Attribution 4.0 International (CC BY 4.0) License used by PLOS journals. Please closely review the details of PLOSu2019s copyright requirements here: PLOS Licenses and Copyright. If you need to request permissions from a copyright holder, you may use PLOS's Copyright Content Permission form.

Potential Copyright Issues:

i) Figure 7A. Please confirm whether you drew the images / clip-art within the figure panels by hand. If you did not draw the images, please provide (a) a link to the source of the images or icons and their license / terms of use; or (b) written permission from the copyright holder to publish the images or icons under our CC BY 4.0 license. Alternatively, you may replace the images with open source alternatives. See these open source resources you may use to replace images / clip-art:

8) We note that your Data Availability Statement is currently as follows: "All data are available in the main text or the supplementary materials.". Please confirm at this time whether or not your submission contains all raw data required to replicate the results of your study. Authors must share the “minimal data set” for their submission. PLOS defines the minimal data set to consist of the data required to replicate all study findings reported in the article, as well as related metadata and methods (https://journals.plos.org/plosone/s/data-availability#loc-minimal-data-set-definition).

9) Please amend your detailed Financial Disclosure statement. This is published with the article. It must therefore be completed in full sentences and contain the exact wording you wish to be published.

1) Please clarify all sources of financial support for your study. List the grants, grant numbers, and organizations that funded your study, including funding received from your institution. Please note that suppliers of material support, including research materials, should be recognized in the Acknowledgements section rather than in the Financial Disclosure

2) State the initials, alongside each funding source, of each author to receive each grant. For example: "This work was supported by the National Institutes of Health (####### to AM; ###### to CJ) and the National Science Foundation (###### to AM)."

3) State what role the funders took in the study. If the funders had no role in your study, please state: "The funders had no role in study design, data collection and analysis, decision to publish, or preparation of the manuscript.".

10) Your current Financial Disclosure states, "The author(s) received no specific funding for this work."

However, your funding information on the submission form indicates seven grants. Please ensure that the funders and grant numbers match between the Financial Disclosure field and the Funding Information tab in your submission form. Note that the funders must be provided in the same order in both places as well. 

Please indicate by return email the full and correct funding information for your study and confirm the order in which funding contributions should appear. Please be sure to indicate whether the funders played any role in the study design, data collection and analysis, decision to publish, or preparation of the manuscript.

**Reviewers' Comments:**

Reviewer's Responses to Questions

**Part I - Summary**

Reviewer #1: (No Response)

Reviewer #2: The manuscript by Zhao et al presents data on the role of fatty acid synthase gene FAS2 in Candida albicans virulence. The FAS2 gene has previously been shown to be essential for systemic as well as oropharyngeal candidiasis. In the present study the authors have attempted to uncover the mechanisms behind avirulent phenotype of fas2�/� null mutant. The authors show that the survival of fas2�/� mutant strain in immunocompetent mice was abolished in cyclophosphamide treated mice. From a series of experiments the authors show that neutrophils are the primary cell type that mediate C. albicans fas2�/� killing. The role of neutrophils in increased fas2�/� killing was further supported by increased b-glucans exposure. Finally, the authors performed experiments to show the vaccine potential of FAS2. Overall, the findings of this work are interesting, and may help better understand the contributions of FAS2 in fungal pathogenesis. The manuscript is well written. The methodology is sound and in most cases the results justify the conclusions.

**Part II – Major Issues: Key Experiments Required for Acceptance**

Reviewer #1: The group did not address the concern from prior review about the use of female mice only.

Reviewer #2: (No Response)

**Part III – Minor Issues: Editorial and Data Presentation Modifications**

Reviewer #1: The group discusses the use of live Candida as a vaccine in the discussion. Prior review recommended that this be edited as the feasibility for a live Candida vaccine is low.

Reviewer #2: 1. Figure 6C, In the Mkc1 and Cek1 phosphorylation western blots the mkc1�/� and cek1�/� null mutant control strains are missing. The tubulin loading control also needs to be repeated.

2. Figure 6E, the quality of Rho1 western blot is very poor.

PLOS authors have the option to publish the peer review history of their article (what does this mean? ). If published, this will include your full peer review and any attached files.

**Do you want your identity to be public for this peer review?** For information about this choice, including consent withdrawal, please see our Privacy Policy .

Reviewer #1: No

Reviewer #2: No

**Figure resubmission:**
---

## [Editor Report · Decision Letter 1]

29 Dec 2024

Dear Dr. Zhao,

We are pleased to inform you that your manuscript 'Systemic infection by Candida albicans  requires FASN-α subunit induced cell wall remodeling to perturb immune response' has been provisionally accepted for publication in PLOS Pathogens.

Best regards,

David R. Andes

Academic Editor

PLOS Pathogens

Michal Olszewski

Section Editor

PLOS Pathogens

Sumita Bhaduri-McIntosh

Editor-in-Chief

PLOS Pathogens

orcid.org/0000-0003-2946-9497

Michael Malim

Editor-in-Chief

PLOS Pathogens

orcid.org/0000-0002-7699-2064

The authors have adequately addressed the queries and requested new information.
---

## [Editor Report · Acceptance letter]

Dear Dr. Zhao,

We are delighted to inform you that your manuscript, "Systemic infection by *Candida albicans*   requires FASN-α subunit induced cell wall remodeling to perturb immune response," has been formally accepted for publication in PLOS Pathogens.

Best regards,

Sumita Bhaduri-McIntosh

Editor-in-Chief

PLOS Pathogens

orcid.org/0000-0003-2946-9497

Michael Malim

Editor-in-Chief

PLOS Pathogens

orcid.org/0000-0002-7699-2064